# The conserved protein adaptors CALM/AP180 and FCHo1/2 cooperatively recruit Eps15 to promote the initiation of clathrin-mediated endocytosis in yeast

**Yidi Sun, Albert Yeam, Jonathan Kuo, Yuichiro Iwamoto, Gean Hu, David G. Drubin** *

Department of Molecular and Cell Biology, University of California, Berkeley, California, United States of America

* drubin@berkeley.edu

## Abstract

Clathrin-mediated endocytosis (CME) is a critical trafficking process that begins when an elaborate endocytic protein network is established at the plasma membrane. Interaction of early endocytic proteins with anionic phospholipids and/or cargo has been suggested to trigger CME initiation. However, the exact mechanism by which CME sites are initiated has not been fully elucidated. In the budding yeast *Saccharomyces cerevisiae*, higher levels of anionic phospholipids and cargo molecules exist in the newly formed daughter cell compared to the levels in the mother cell during polarized growth. Taking advantage of this asymmetry, we quantitatively compared CME proteins in *S. cerevisiae* mother versus daughter cells, observing differences in the dynamics and composition of key endocytic proteins. Our results show that CME site initiation occurs preferentially on regions of the plasma membrane with a relatively higher density of endocytic cargo and/or acidic phospholipids. Furthermore, our combined live cell-imaging and yeast genetics analysis provided evidence for a molecular mechanism in which CME sites are initiated when Yap1801 and Yap1802 (yeast CALM/AP180) and Syp1 (yeast FCHo1/2) coordinate with anionic phospholipids and cargo molecules to trigger Ede1 (yeast Eps15)-centric CME initiation complex assembly at the plasma membrane.

## Introduction

Clathrin-mediated endocytosis (CME) is a key cellular pathway in eukaryotes that mediates nutrient uptake, regulates response to extracellular stimuli, and controls the plasma membrane's chemical composition and surface area. From yeast to humans, cells recruit several dozen highly conserved proteins to CME sites in a sequential manner over a period of approximately 2 min to facilitate cargo capture and plasma membrane invagination (Fig 1A, left panel) [1–3]. Since clathrin has neither lipid- (membrane) nor cargo-binding ability on its own, several early arriving adaptor and accessory proteins, such as AP-2, FCHo1/2 (Syp1 in

**Data Availability Statement:** All relevant data are within the paper and its Supporting Information files.

**Funding:** This work was supported by National Institutes of Health grant R35GM118149 (D.G.D). The funders had no role in study design, data collection and analysis, decision to publish, or preparation of the manuscript.

**Competing interests:** The authors have declared that no competing interests exist.

**Abbreviations:** CME, clathrin-mediated endocytosis; MIP, Maximum Intensity Projection; PCC, Pearson correlation coefficient; PS, phosphatidylserine; ROI, regions of interest; UIM, ubiquitin-interacting motif.

yeast), and Eps15 (Ede1 in yeast), are suggested to recruit clathrin to nascent CME sites [4–12]. Due to inconsistent results between studies, the importance of each early-arriving protein for CME initiation has remained elusive [4–6,13–17].

In recent years, increasing evidence supports a central role for the early acting endocytic protein Eps15 (Ede1 in yeast) in CME site initiation, possibly through the formation of a liquid-like condensate [18–22]. Knocking down both Eps15 and its homolog Eps15R in mammalian cells resulted in a 40% inhibition of transferrin and EGF uptake [23]. Similarly, deletion of *EDE1* in yeast resulted in an overall 50% decrease in the number of productive endocytic events, and partial inhibition of the internalization step of CME [11,24]. However, neither Eps15 in mammals nor Ede1 in yeast has known lipid membrane-binding ability [21,25]. Thus, to understand how CME is initiated, it is crucial to determine how Eps15 and Ede1 interact with various other early-arriving endocytic proteins, potentially in complexes with liquid-like properties, to trigger Eps15 (Ede1)-centric initiation complex assembly at the plasma membrane.

In the budding yeast *Saccharomyces cerevisiae*, actin assembles at CME sites at the late stage of CME to facilitate membrane invagination and vesicle scission (Fig 1A, left panel) [26]. In addition, the interval from the appearance of the earliest arriving endocytic proteins (such as Ede1) to the beginning of actin assembly at a CME site ranges from 30 s to a few min (Fig 1A, left panel) [11]. Interestingly, actin patches, now known to be late-stage CME sites, were shown to be much more concentrated in the daughter cell than the mother cell in yeast undergoing polarized bud growth (Fig 1A, right panel) [27,28]. Thus, the asymmetric distribution of actin patches in these cells implies different rates of endocytic site initiation and/or site maturation between the mother and the daughter cell, which share the same pool of endocytic proteins. Therefore, a detailed comparative analysis of CME site initiation and maturation between these 2 compartments could potentially reveal clues into their mechanisms.

Cell morphology, spatial control of exocytosis, and cell cycle progression are all inextricably intertwined in budding yeast [29–31]. As a daughter cell first emerges from a mother cell as a nascent bud, exocytosis is spatially constrained to the region of the daughter cell surface undergoing rapid expansion, resulting in more endocytic cargos such as SNARE proteins accumulating on the growing daughter surface (reviewed by [32]). The anionic phospholipids phosphatidylserine (PS) and phosphatidylinositol-4,5-bisphosphate (PI(4,5)P$_2$) play important roles in cell polarity development and endocytosis [33–39] and are observed to be enriched at the bud cell surface, the bud neck, and the tips of mating projections, which are all areas active in exocytosis and enriched for late-stage endocytic sites [36,40–42]. Thus, both cargo molecules and anionic phospholipids exhibit asymmetric distribution on the cortex of budding yeast undergoing polarized bud or mating projection growth, with higher levels in the growing daughter (the bud) or mating projection than in the mother. Therefore, a yeast cell offers a unique opportunity to compare dynamics of CME site initiation and maturation at plasma membrane regions with relatively high (daughter cells) or low (mother cells) levels of phospholipids and cargo molecules.

In this study, we quantitatively compared behaviors of endocytic proteins in *S. cerevisiae* mother and daughter cells undergoing polarized growth, demonstrating that the dynamics and composition of key endocytic proteins at CME sites, as well as the frequency of CME events, differ between the 2 compartments. These results further established that CME site initiation occurs most frequently in regions of the cell surface with relatively higher density of anionic phospholipid and cargo. Importantly, live-cell imaging of yeast mutants revealed critical roles for paralogs Yap1801 and Yap1802 (yeast homologs of mammalian CALM/AP180) and Syp1 in promoting cortical Ede1 recruitment and assembly at CME sites to initiate the process by coordinating with anionic phospholipids and possibly cargo through their multifunctional interaction domains. The presence of lipid- and cargo-binding domains in various endocytic proteins allows for plasticity in how CME sites are initiated.

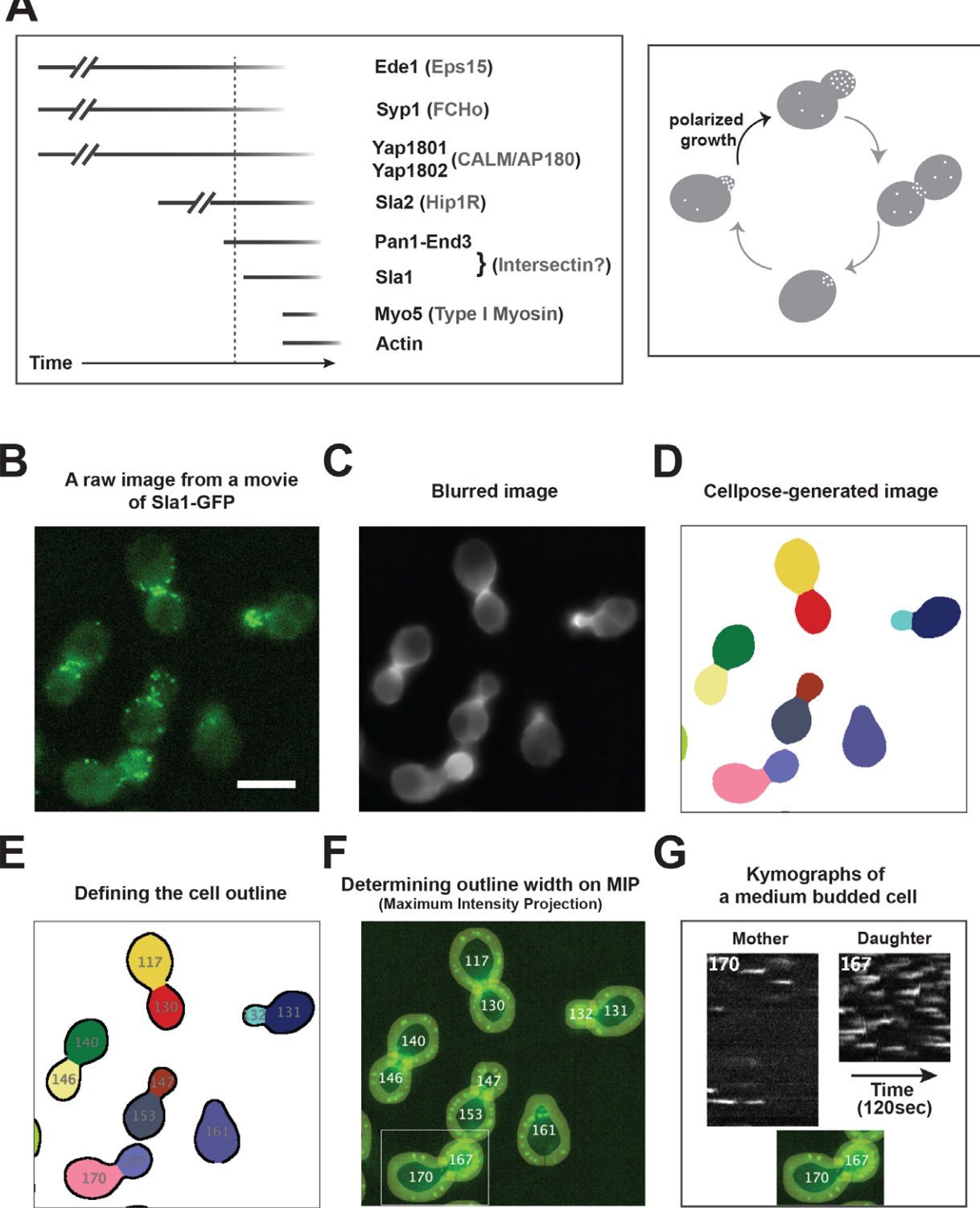

**Fig 1. Protein recruitment dynamics at yeast CME sites and an automated strategy to generate separate circumferential kymographs for mother and daughter cells from a movie of budded yeast cells.** (A) Previously determined timeline for the presence of a subset of endocytic proteins at CME sites (left) [1]. The timeline can be divided into 2 phases separated by the vertical dotted line: the early phase on the left and the late phase on the right. Pan1 and End3 link the 2 phases [50,51]. The mammalian homologs of the yeast proteins are indicated in parentheses. The Pan1-End3-Sla1 complex contains similar interaction domains to the mammalian endocytic protein intersectin and is proposed to perform a similar

function [53]. The length of the early phase is variable (30 s to 2 min), indicated by "//", while the length of the late phase is much more regular (approximately 30 s). AP-2, Pal1, Pal2, and clathrin exhibit similar dynamics to the Yap1801 and Yap1802 [24]. Actin patch (white dot) localization in budding yeast at different cell cycle stages (right) [27,28]. The cells with small or medium buds are undergoing polarized cell growth. (B–G) Show the workflow we developed to quantify protein lifetimes at CME sites. A detailed description is provided in the Results and Materials and methods sections. The scale bar is 5 μm. CME, clathrin-mediated endocytosis.

## Results

### Automated circumferential kymograph generation for analysis of live-cell imaging data

To comprehensively and quantitatively compare the dynamics and lifetimes of proteins at CME sites in mother versus daughter cells of *S. cerevisiae* undergoing polarized bud growth, we developed an automated process to generate separate circumferential kymographs [43] for the 2 compartments of a budded cell, the mother cell and the daughter cell (Fig 1B–1G). We used 2D wide-field fluorescence microscopy at the equatorial focal plane of yeast cells to record the dynamics of endocytic proteins tagged with fluorescent proteins. Each fluorescently labeled protein was expressed from its own promoter at its native chromosomal locus. Previous studies demonstrated that endocytic proteins are visible at the equatorial focal plane for nearly their entire lifetime at each detected CME site [26,44,45].

In our approach, a raw image from a movie (Fig 1B) is first blurred (Fig 1C) using the median filter and erosion function in ImageJ software (https://imagej.nih.gov/ij/). The use of a blurred image makes it easier for Cellpose to segment yeast cells precisely for subsequent analysis. Cellpose is a recently developed, deep learning-based segmentation algorithm in Python 3 [46]. In the Cellpose-segmented images (Fig 1D), budded cells were recognized as 2 objects: mother and daughter cells. We developed an ImageJ plugin to define the outlines of each cell segmented by Cellpose. As shown in Fig 1E, outlines of 2 objects from a budded cell were assigned as separate regions of interest (ROI) with identifiers (numbers). The bud neck regions were excluded from the analysis because the CME sites at the neck regions are too crowded to allow individual CME sites to be identified. The width of the outline was adjusted to cover all the fluorescence signals present on the cell cortex in the movie's Maximum Intensity Projection (MIP) (Fig 1F). The cortical fluorescence signals on the MIP represent all of the endocytic events that occurred during the time interval captured by the movie. Finally, another custom-made ImageJ plugin automatically generated circumferential kymographs along all the defined outlines [47]. For example, 2 separate kymographs, one for the mother cell and one for the daughter cell, were shown for a budded cell (Fig 1G). In the circumferential kymographs, each short white trace represents the progression of the appearance, accumulation, and disappearance of a particular fluorescently labeled protein, in this case, Sla1-GFP, at a CME site (Fig 1G). Using this strategy, all endocytic events that occur on the cell cortex of either the daughter or mother cell (except in the neck region) at the equatorial focal plane over a given time interval can be visualized and quantified from a 2D representation in the form of a circumferential kymograph.

### The frequency, lifetime, and lifetime regularity of key endocytic proteins differ at CME sites in mother and daughter cells of budding yeast undergoing polarized growth

The type I myosin Myo5 appears at CME sites at approximately the same time as the onset of actin polymerization and aids endocytic membrane invagination (Fig 1A, left panel) [44,48,49]. Myo5-GFP patch dynamics were compared between the mother and daughter cells of yeast undergoing polarized growth using circumferential kymographs generated from 60-s

movies (Fig 2A–2C). The Myo5-GFP traces mark each productive endocytic event, and the trace length represents the Myo5 lifetime at a CME site (Fig 2A). In mother and daughter cells, Myo5 traces exhibited similar average lifetimes (approximately 9 s) (Fig 2B) and coefficients of variation (approximately 18%). However, Myo5 traces along the cell cortex at the equatorial focal plane were approximately 7 times more abundant in daughter cells compared to mother cells (Fig 2C). Together, these data suggest that productive CME events occur much more frequently in daughter cells compared to mother cells.

To gain insights into how CME sites initiate and mature differently in daughter cells vs. mother cells, we examined the lifetime of several highly conserved key endocytic proteins that were previously reported to arrive before Myo5 (Fig 2D and 2E). Sla1 is an adaptor protein containing 3 SH3 domains that regulates the initiation of endocytic actin filament assembly. The average lifetimes for Sla1 in the mother were slightly longer than for the daughter (approximately 25 s vs. approximately 28 s) (Fig 2D and 2E). Like Myo5, the coefficient of variation for Sla1 lifetime in either of the compartments was close to 20% (Fig 2F), establishing that Sla1 lifetimes are also fairly regular.

In contrast to Myo5 and Sla1, Sla2 (Hip1R in mammalian cells) in mother cells exhibited lifetimes that were longer than in daughter cells by several fold or more (Fig 2D and 2E). Some Sla2 patches in the mother cell persisted for the entire 3-min movie (Fig 2D and 2E). Therefore, the true Sla2 lifetimes in mother cells were underestimated. However, the average lifetimes for Sla2 in daughter cells were around 26 s (Fig 2E), and the coefficient of variation for the lifetimes was close to 20% (Fig 2F). Thus, in terms of lifetime length and regularity, Sla2 exhibits very different dynamics in the mother versus daughter cells of budding yeast undergoing polarized growth.

Pan1 is a key protein linking actin assembly to the CME site and is essential for cell growth [50–52]. The Pan1-End3-Sla1 complex is proposed to function similarly to mammalian intersectin, a single protein that contains many of the same interaction domains found collectively in Pan1, End3, and Sla1 [53]. Circumferential kymographs revealed a previously undescribed behavior of Pan1 in mother cells in that Pan1 displays a low-intensity stage followed by a high-intensity stage (Fig 2D). Intriguingly, the duration of the Pan1-GFP high-intensity stage in the mother cells was comparable to the average Pan1 lifetime in daughter cells (approximately 26 s versus approximately 25 s, respectively), while the average lifetime of the combined low-intensity plus high-intensity stages in the mother cells was more than 60 s (Fig 2E).

While the lifetime of late-arriving protein Myo5 is relatively regular and approximately the same in the mother and daughter cells, endocytic event completion (indicated by Myo5 recruitment) occurs several-fold more frequently in daughter cells compared to mother cells. On the other hand, the lifetimes of Sla2 and Pan1, recruited about halfway through the CME pathway (Fig 1A, left panel), are longer and have greater variation in the mother cell than in the daughter cell (Fig 2D–2F). Together, these observations indicate that CME sites mature faster and more regularly in the daughter cells compared to mother cells. Furthermore, these data suggest that early-arriving proteins (i.e., those appearing before Sla2 and Pan1) likely also behave differently between mother and daughter cells, which we next examined.

## The composition of early-arriving endocytic proteins at CME sites differs between mother and daughter cells of yeast undergoing polarized growth

To gain insights into factors involved in CME site initiation, we performed two-color live-cell imaging of mother cells and daughter cells in yeast undergoing polarized growth, directly comparing the dynamics of known early-arriving endocytic proteins with those of proteins that appear at the later stages of the pathway.

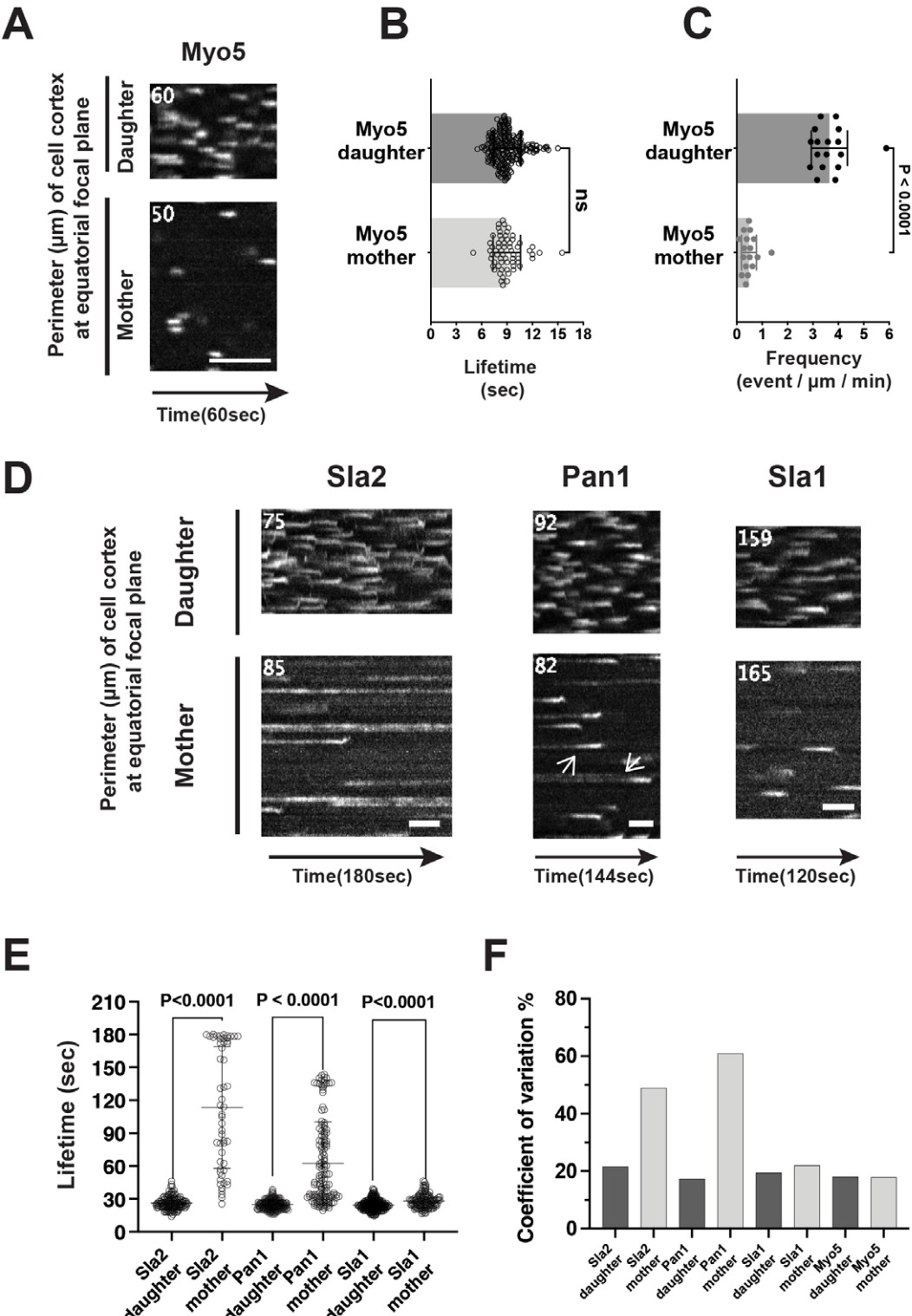

**Fig 2. Frequency and dynamics of key endocytic proteins at CME sites in mother vs. daughter cells of budding yeast undergoing polarized growth.** (A) Representative circumferential kymographs of Myo5-GFP in mother and daughter cells of yeast cells undergoing polarized growth. (B, C) Myo5-GFP patch lifetime and frequency in mother vs. daughter cells. The numerical data are presented in S1 Data. (D) Representative circumferential kymographs of indicated GFP-tagged proteins in mother and daughter cells. Note: the kymograph of Pan1 in the mother cell reveals 2 stages, a low abundance stage followed by a

high abundance stage, with transition points between the stages indicated by arrows. The kymograph numbers identify different ROIs for the mother and daughter cells. (E, F) Mean with SD and coefficients of variation for the lifetimes of the indicated proteins in mother and daughter cells. The numerical data are presented in S1 Data. The scale bar is 30 s. CME, clathrin-mediated endocytosis; ROI, regions of interest; SD, standard deviation.

In daughter cells, Ede1 patches assembled at the cell cortex slightly before Sla2 and disassembled before Sla2 disappeared (Fig 3A and 3B). Ede1 disappeared when Myo5 began to accumulate at CME sites (Fig 3C). The lifetime of Ede1 in the daughter cell was short and regular (31.7 ± 6.9 s with a coefficient of variation of 23%) (Fig 3D). However, in the mother cell, Ede1 exhibited much longer and more variable lifetimes (95.5 ± 47.2 s lifetime with a coefficient of variation of 48%) (Figs 3A–3D and 4B). Like Sla2, some Ede1 patches in the mother cell persisted for the entire duration of 3-min movies, and some Ede1 patches appeared during the movie and persisted through the movie's end (Fig 3A–3C). Therefore, Ede1 lifetimes in the mother cell were underestimated. More importantly, Ede1 sites formed approximately 5 times more frequently in daughter versus mother cells (Fig 3D). Since Ede1 is one of the earliest appearing proteins at CME sites, our results demonstrate that endocytic site initiation occurs much more frequently in daughter cells compared to mother cells in budding yeast undergoing polarized growth.

We also analyzed the dynamics and localization of another early-appearing endocytic protein, Yap1801 (CALM (clathrin assembly lymphoid myeloid leukemia protein)/AP180 in mammals), and its paralog, Yap1802. Yap1801 and Yap1802, which are hereafter referred to collectively as the Yap180s, were both tagged with GFP in the same strain and their dynamics were compared to those of Ede1-mScarlet-I because the paralogs appear to fulfill the same function (Fig 3F) [54]. In daughter cells, Yap180s appeared around the same time as Ede1, and they persisted slightly longer than Ede1 (Fig 3F). Consistently, the Yap180s' GFP signal at endocytic sites lasted until Myo5 disappeared (Fig 3E), while Ede1 disappeared before Myo5 appeared (Fig 3C). The average lifetime of the Yap180s in daughter cells was around 36 s, with a coefficient of variation of 19%. However, compared to daughter cells, the Yap180s' signal was noticeably dimmer in mother cells (Fig 3E and 3F). In addition, while almost all of the Myo5 sites in mother cells (96.5%, $n = 164$) were initiated with Ede1, only around 83% ($n = 133$) of Myo5 sites in mother cells were initiated with Yap180s, indicating that nearly 20% of CME sites in mother cells do not contain detectable Yap1801 or Yap1802. Since Yap1802-GFP appeared to be brighter and more strongly polarized than Yap1801-GFP (S1 Fig), we further compared Yap1802-GFP and Myo5-mScarlet-I in mother cells. Strikingly, 87% ($n = 154$) of the Myo5 traces did not align with Yap1802-GFP signal in mother cells (Fig 3G), indicating that most CME sites in mother cells lack detectable Yap1802.

Syp1 (FCHo1/2 in mammals) was previously suggested to form a complex with Ede1 in the cytoplasm, with this complex subsequently associating with CME sites on the plasma membrane [55]. Indeed, kymographs revealed that these 2 proteins arrive at CME sites with very similar timing in both mother and daughter cells (Fig 3H). In addition, the lifetime of Syp1 in mother and daughter cells was 91.7 ± 51.8 s and 30.4 ± 5.8 s, respectively. Two other early arriving proteins, Apl1 (one of the AP-2 subunits) and Pal1 (with no known mammalian homolog), respectively, were much dimmer at, or apparently absent from, most CME sites in mother cells (S1 Fig).

The results thus far are summarized in Fig 4. The lifetimes of early- and middle-arriving proteins in daughter cells were much shorter (around 30 to 40 s) and more regular (with coefficient of variations close to 20%) compared to previously reported lifetimes from studies in which CME dynamics data were averaged for events in daughter and mother cells (Fig 1A, left

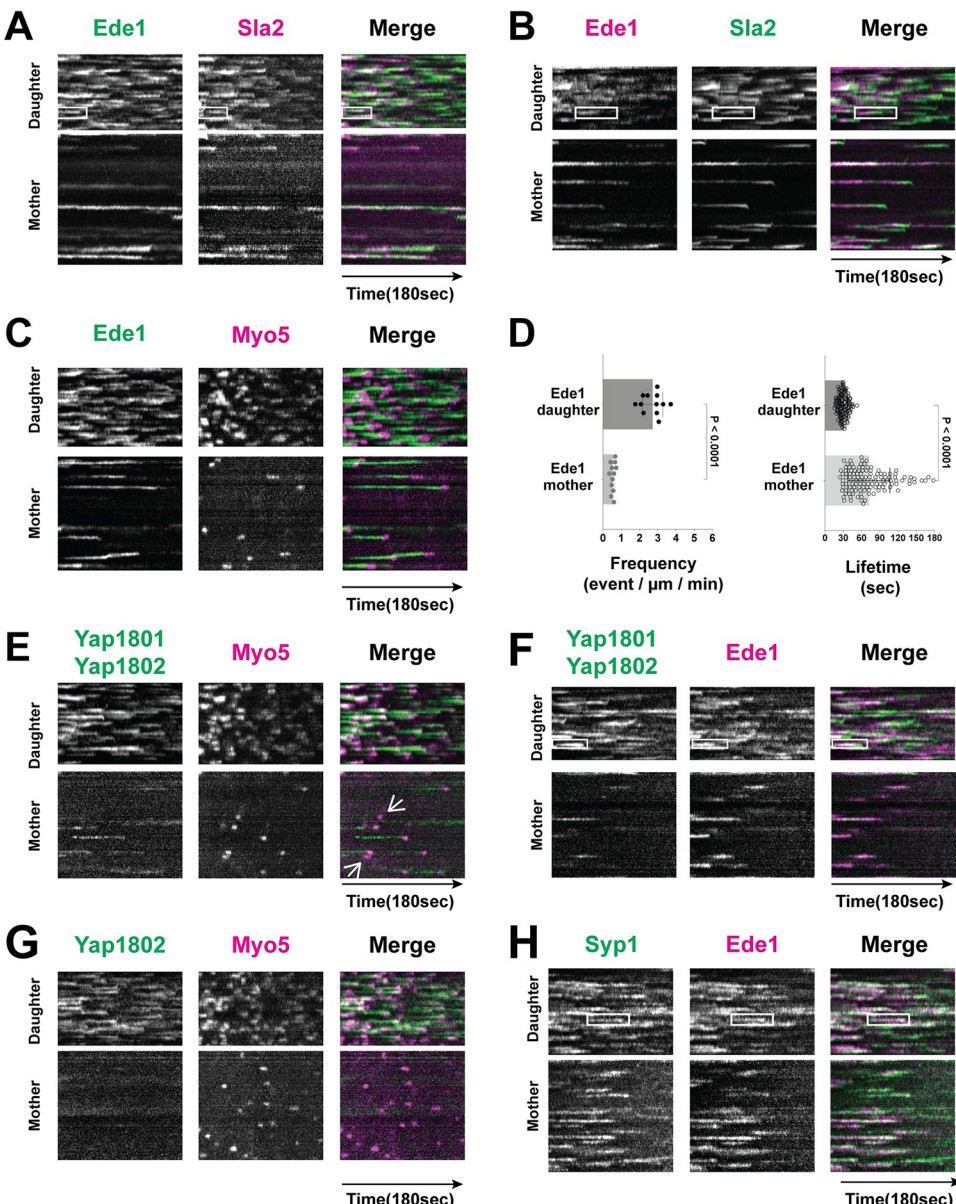

**Fig 3. The composition of early-arriving endocytic proteins at CME sites differs between mother and daughter cells of budding yeast undergoing polarized growth.** (A–C and E–H) Two-color circumferential kymograph representations of GFP- or mScarlet-I-tagged proteins in mother and daughter cells. The boxes on the kymographs from daughter cells show the dynamic behavior of the indicated protein pair at CME sites. (D) Ede1-GFP patch lifetime and frequency of appearance in mother vs. daughter cells. The numerical data are presented in S2 Data. (E) Some endocytic sites in mother cells lack detectable Yap1801 and Yap1802. The arrows in the mother cell kymograph show productive CME sites labeled by Myo5 but lacking Yap1801 and Yap1802. (G) Most (87%) Myo5 sites in mother cells lack Yap1802-GFP. CME, clathrin-mediated endocytosis.

panel). Early- and middle-arriving proteins that displayed longer and more variable lifetimes were mainly observed in the mother cells for yeast undergoing polarized bud growth, consistent with previous observations [56]. On the other hand, the length of the actin assembly phase (represented by Myo5) was identical between mother and daughter cells of polarized yeast. Together, these results indicate that the time from CME site initiation to

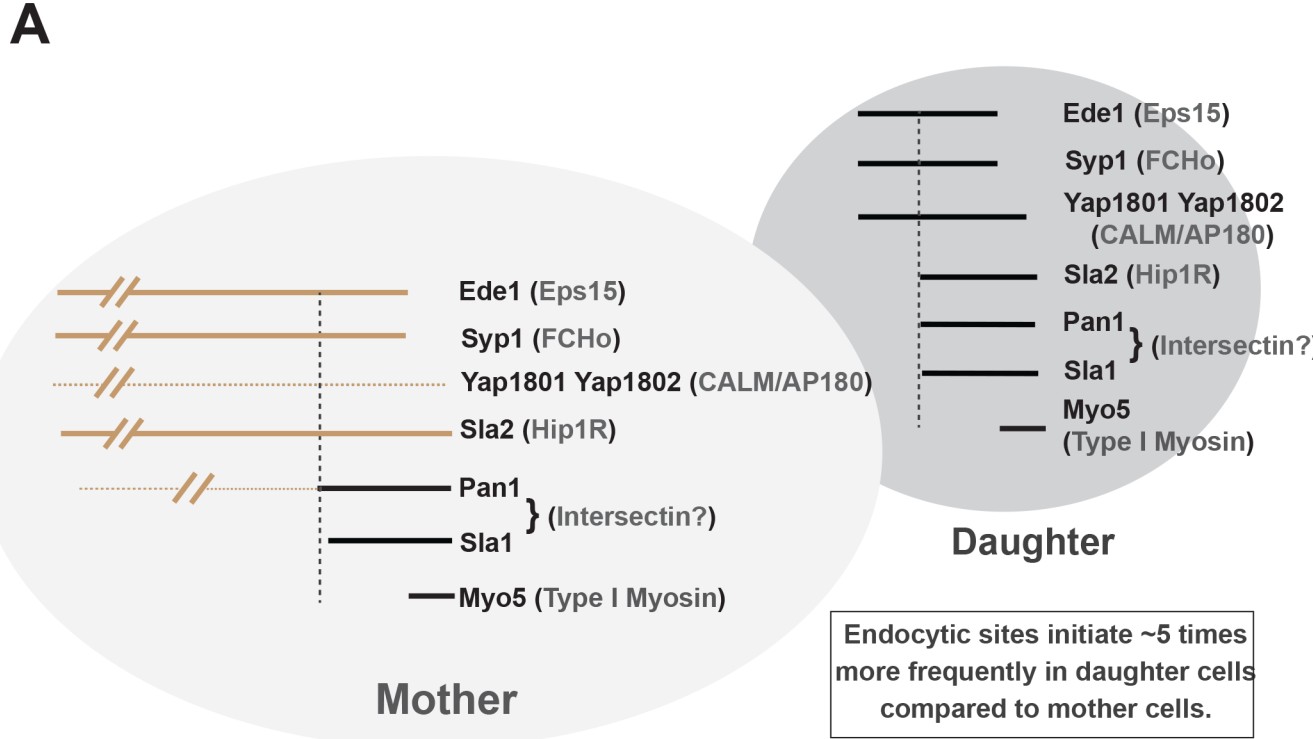

**B**

| Protein | Yap1802 | Ede1 | Syp1 | Sla2 | Pan1 | Sla1 | Myo5 |
|---|---|---|---|---|---|---|---|
| Lifetime in daugher | 36.3 ± 6.7 (n=110) | 31.7 ±6.9 (n=100) | 30.4 ±5.9 (n=109) | 26.1 ± 5.6 (n=135) | 24.8 ± 4.3 (n=289) | 24.5 ± 4.7 (n=233) | 9.0 ± 1.6 (n=216) |
| Lifetime in mother | N.D. | 95.5¶ ±47.2 (n=39) | 91.7¶ ±51.8 (n=59) | 113.5¶ ± 55.6 (n=54) | 62.4¶ § ± 38.0 (n=137) | 28.1 ± 6.2 (n=118) | 9.0 ±1.6 (n=63) |

**Fig 4. Comparison of CME site initiation in mother vs. daughter cells of yeast undergoing polarized bud growth.** (A) The lifetimes of early- and mid-arriving endocytic proteins are much shorter and less variable in the daughter cells (black lines) than in the mother cells (brown lines). In contrast, Sla1 and Myo5 exhibit similar, regular lifetimes in mother and daughter cells. Yap1801 and Yap1802 signals are noticeably dimmer in mother cells (dotted line), and most CME sites in mother cells lack detectable Yap1802. The Pan1 signal in mother cells can be divided into a low-abundance stage (dotted brown line) followed by a high-abundance stage (black line). The length of Pan1's high abundance phase in mother cells is similar to Pan1's lifetime in daughter cells. CME sites (represented by Ede1) initiate approximately 5 times more frequently in daughter cells compared to the mothers, correcting for differences in the surface distance sampled. The different shades of gray color are intended to signify that cargo and anionic phospholipids exhibit asymmetric distribution, with higher levels on the plasma membrane of daughter cells in small to medium budded cells. (B) Summary of protein lifetimes (mean with standard deviation (sec)) at CME sites in mother and daughter cells of yeast undergoing polarized bud growth. ND means not determined. ¶ The true lifetimes should be greater than the indicated values because some events are not completed within the duration of our movies. § Lifetime represents the low-intensity and high-intensity stages of Pan1 combined. The numerical data are presented in S1 Data and S3 Data. CME, clathrin-mediated endocytosis.

actin assembly is much longer in mother cells than in daughter cells. Unexpectedly, we observed that most CME sites in mother cells lack detectable levels of the early-arriving proteins Yap1802 and Pal1. This polarized localization of Yap1802 and Pal1 correlates well with the asymmetric distribution of anionic phospholipids and cargo, both of which are enriched in daughter cells relative to mother cells. Yap180s carry lipid-binding and ubiquitin (cargo) binding motifs [57–59], which leads us to speculate that higher levels of anionic phospholipids and cargo in the daughter cell might recruit early-arriving endocytic proteins, in turn controlling CME site initiation. Our observation that CME site initiation occurred at a much higher frequency in daughter cells of polarized yeast compared to mother cells is in good agreement with this possibility. We sought to test this hypothesis by investigating how early endocytic proteins are recruited to the plasma membrane of budded yeast cells.

## Polarized enrichment of the Yap180s and Syp1 in daughter cells is independent of Ede1

Ede1 is thought to be a key factor in CME site initiation [11]. The Ede1 N-terminus interacts with the Yap180s and Pals (Pal1 and Pal2) through an EH-NPF domain interaction (Fig 5A) [60,61], while the Ede1 C-terminus interacts with the μ-Homology domain of Syp1 (Fig 5A) [12]. While AP-2 was shown to associate with Eps15 constitutively in mammals [62], a direct interaction between AP-2 and Ede1 has not been documented in yeast. The Yap180s and Syp1 can bind to both lipids and cargo [12,57,59], while Ede1 has not been reported to bind to lipids.

We next addressed whether Ede1 affects the recruitment of the Yap180s or Syp1 to the cell cortex. We imaged Syp1-GFP alone or Yap1801-GFP and Yap1802-GFP together, over time, in wild-type or *ede1Δ* cells (Fig 5B and 5C). MIPs were generated from movies to provide a complete view of the localization of each GFP-tagged endocytic protein during the 3-min duration of the movie. In the wild-type cells, Syp1-GFP, or Yap1802-GFP and Yap1801-GFP, were observed along the entire cell cortex, but highly concentrated in daughter cells. In *ede1Δ* cells, Syp1-GFP or Yap1801-GFP and Yap1802-GFP maintained polarized enrichment in daughter cortex (Fig 5B and 5C). However, compared to wild-type cells, both Syp1-GFP and Yap1801-GFP and Yap1802-GFP cortical signals were more continuous (less punctate) in *ede1Δ* cells. Thus, without Ede1, both Syp1-GFP and Yap1801-GFP and Yap1802-GFP are still recruited to the cell cortex in a polarized manner, but their ability to coalesce at CME sites is reduced. Importantly, Syp1 1-565a.a.-GFP and Yap1802 1-339a.a.-GFP, which retain N-terminal lipid-binding domains (EFC/F-BAR domain of Syp1 and ANTH domain of Yap1802) but lack their Ede1-binding regions, still showed cortical, polarized enrichment in daughter cells but to a lesser extent than the corresponding full-length proteins (Fig 5B and 5C). Additionally, Syp1's EFC/F-BAR domain and Yap1802's ANTH domain, respectively, were required for Syp1 and Yap1802 cortical localization ([12] and S2 Fig). These results demonstrate that the cortical polarization of Syp1 and Yap1802-GFP are independent of Ede1 and instead depend on the lipid-binding domains of these proteins.

We also examined the localization of 2 other early-arriving endocytic proteins, Apl1 (one of the 4 subunits of AP-2) and Pal1, in *ede1Δ* cells (S2 Fig). AP-2 cortical localization was mostly lost, while Pal1 cortical localization was not affected in *ede1Δ* cells. Our results are consistent with previous observations that Ede1 is required for cortical recruitment of AP-2, but not for Pal1/2 [24,61].

In summary, cortical polarized localization of Yap1802, Yap1801, Syp1, and Pal1 is independent of Ede1, while AP-2 cortical recruitment requires Ede1.

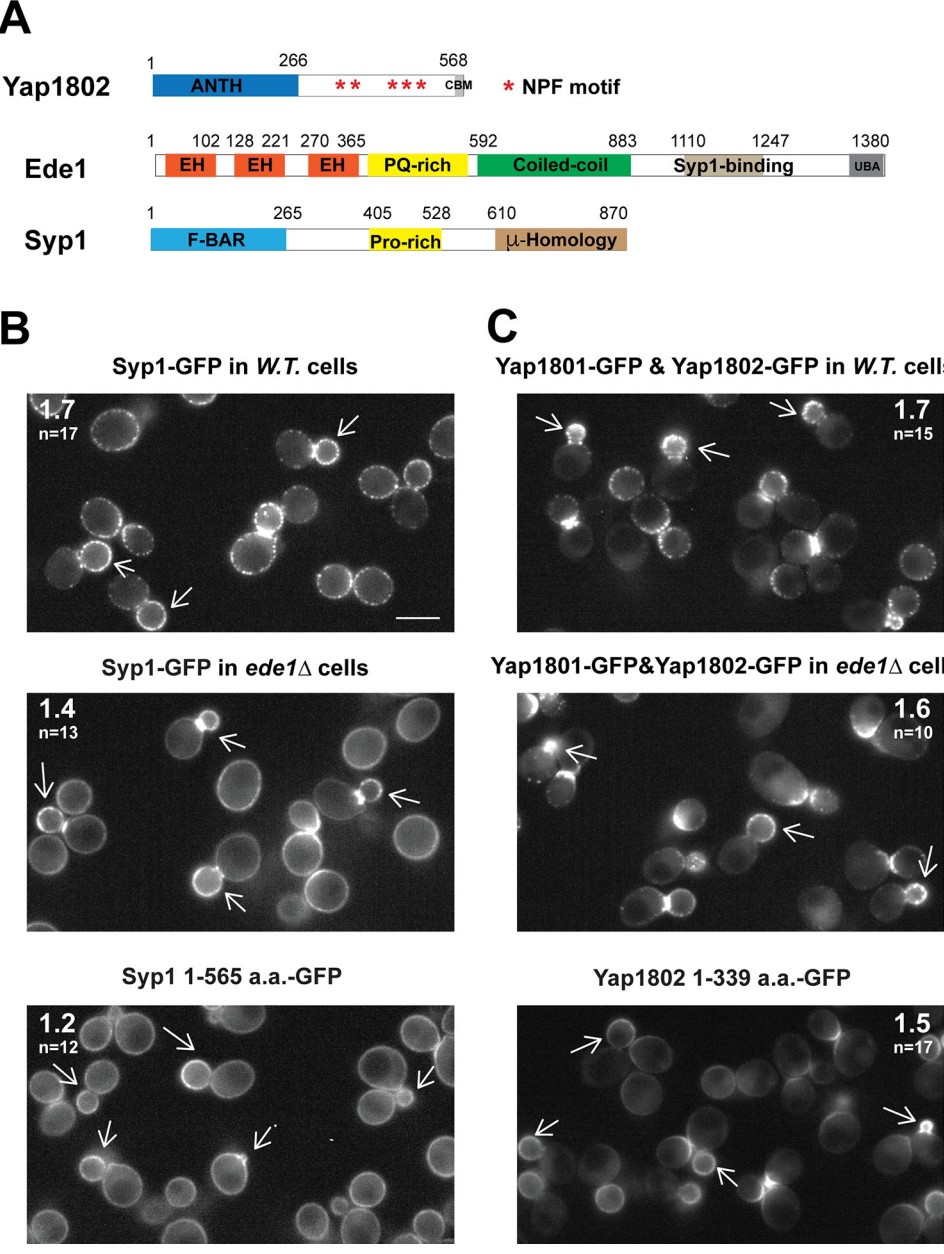

**Fig 5. Polarized enrichment of Yap1801, Yap1802, and Syp1 at the daughter cell plasma membrane is independent of Ede1.** (A) Schematic diagrams of domain structures in Yap1802, Ede1, and Syp1. ANTH domain, AP180 N-terminal homology domain; NPF, Asn-Pro-Phe motif; CBM, clathrin box motif; EH, Eps15 homology; PQ-rich, proline and glutamine-rich motif; UBA, Ubiquitin-Associated domain; F-BAR, FCH-bin-amphiphysin-Rvs domain. (B, C) MIP from 3 min movies taken from indicated yeast strains. The arrows point to the enriched localization of the GFP-labeled proteins in the daughter cells of yeast with small- to medium-sized daughters. The average ratio (daughter vs. mother) of GFP signal intensity along the cortex of polarized cells is indicated in the upper left or upper right corner of images in (B) or (C), respectively. The numerical data are presented in S4 Data. The scale bar is 5 μm. MIP, Maximum Intensity Projection.

## Ede1 is mis-localized in cells lacking certain early-arriving proteins or protein combinations

A previous study reported that in cells lacking Yap1801 and Yap1802 as well as the AP-2 adaptor subunit Apl3 (*yap1801Δ yap1802Δ apl3Δ*), Ede1 only weakly localizes to the cell cortex

[55]. In these cells, the excess cytosolic Ede1 assembled into large cytoplasmic puncta with liquid-like properties [19]. These observations suggest that Yap180s and AP-2 might be involved in tethering Ede1 to the cell cortex. To more thoroughly examine how Ede1 is recruited to the cell cortex, we explored Ede1 localization in cells lacking various combinations of early-arriving proteins.

Ede1 cortical localization was largely normal, and cytoplasmic Ede1 puncta appeared at a very low frequency (approximately 3%, *n* = 358) in *apl1Δ* cells, similar to wild-type cells (approximately 3%, *n* = 172) (Fig 6A and 6B). Apl1 and Apl3 are two of the 4 subunits of the hetero-tetrameric AP-2 complex. Results similar to those obtained with *apl1Δ* cells were obtained in *apl3Δ* cells. Thus, AP-2 absence alone did not appreciably affect Ede1 recruitment. However, cytoplasmic Ede1 puncta were observed in 26% (*n* = 384) of the *syp1Δ* cells and 32% (*n* = 233) of the *yap1801Δ yap1802Δ* cells, indicating that Ede1 plasma membrane recruitment was partially impaired in these mutants (Fig 6C and 6D). Lacking either Yap1801 or Yap1802 caused 11% (*n* = 105) or 15% (*n* = 214), respectively, of cells to accumulate cytosolic Ede1 puncta (S3 Fig).

More than 90% of *apl1Δ syp1Δ* (*n* = 192) or *apl1Δ yap1801Δ yap1802Δ* (*n* = 222) cells contained cytoplasmic Ede1 puncta (Fig 6E and 6F). In these mutant cells, however, cortical Ede1 was still visible (Fig 6E and 6F). In contrast, Ede1 was mostly detected in cytoplasmic puncta rather than at the cell cortex in *syp1Δ yap1801Δ yap1802Δ* (*n* = 145), or *syp1Δ yap1801Δ yap1802Δ apl1Δ* cells (*n* = 235) (Fig 6G and 6H). Consistently, the cytoplasmic Ede1 puncta in *syp1Δ yap1801Δ yap1802Δ*, or *syp1Δ yap1801Δ yap1802Δ apl1Δ* cells appeared larger and brighter than those in *apl1Δ syp1Δ* or *apl1Δ yap1801Δ yap1802Δ* cells (Fig 6).

Our results indicate that Syp1 and the Yap180s play crucial roles in Ede1 recruitment to the cell cortex. AP-2 only plays a critical role in Ede1's cortical recruitment when the Yap180s and/or Syp1 is/are also absent.

## Ede1's EH motifs and Syp1-binding domain are required for its cortical localization

To further illustrate the molecular mechanisms that underlie Ede1's cortical recruitment to CME sites, we examined the localization of several Ede1 mutants lacking motif(s) that mediate interactions with other early-arriving proteins. All *ede1* mutants were integrated into the endogenous *EDE1* locus, replacing the wild-type gene.

Previous studies suggested that the 3 Eps15 homology (EH) domains of Ede1 interact with the Yap180s and Pal1 and Pal2 through their NPF (Asn-Pro-Phe) domains [60,61]. *ede1 EH\** is a mutant in which only the conserved tryptophan residues in Ede1's EH domains were mutated to alanine (W56A, W176A, W319A) (Fig 7A) [63]. Mutation of these conserved tryptophan residues dramatically impairs the binding of the EH domains to NPF motifs [64]. Strikingly, in 93% (*n* = 223) of cells expressing *ede1 EH\*-GFP*, the mutant protein appeared in cytoplasmic puncta (Fig 7B). The Ede1 puncta formation phenotype in *ede1 EH\*-GFP* cells was much more severe than in *yap1801Δ yap1802Δ* cells (Fig 6D). However, 88% of cells (*n* = 162) lacking the 4 proteins Yap1801, Yap1802, Pal1, and Pal2, showed cytosolic Ede1 puncta (S3 Fig). These results suggest that Pal1 and Pal2 might be redundant with Yap1801 and Yap1802 in mediation of Ede1 localization.

The *ede1-Δsyp1-binding* mutant lacks amino acids 1,100 to 1,247, the region previously shown to interact with Syp1 (Fig 7A) [12], and 15% (*n* = 221) of *ede1-Δsyp1-binding* cells showed cytoplasmic puncta (Fig 7C), phenocopying Ede1 localization in *syp1Δ* cells (Fig 6C).

Besides cytoplasmic puncta, ede1 EH\*-GFP and ede1-Δsyp1 binding-GFP were also observed at the cell cortex (Fig 7B and 7C). However, strikingly, when both domains were

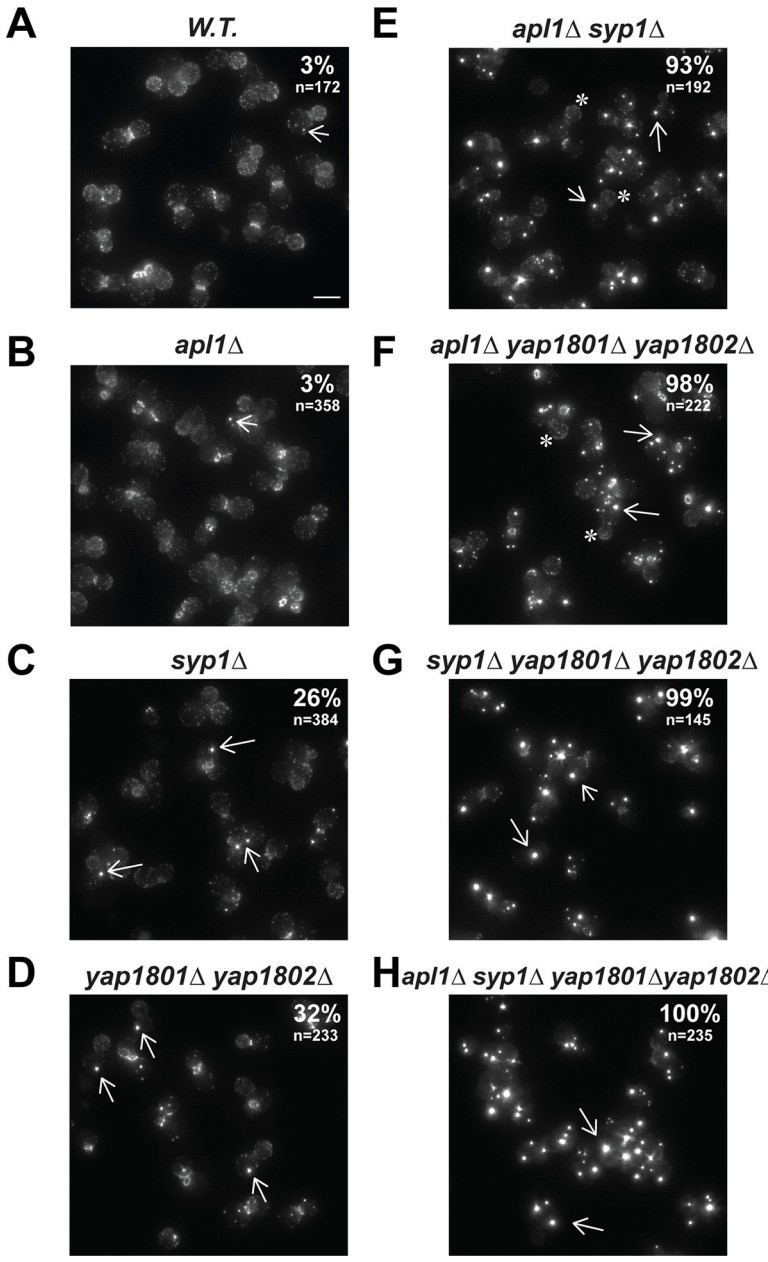

**Fig 6. Cytoplasmic Ede1 puncta appear in mutants lacking various early-arriving endocytic proteins.** Maximum Z-projections of Ede1 localization in the indicated null mutants. The number in the upper right corner in each panel indicates the percentage of cells that contain cytoplasmic Ede1 puncta. *n* equals the number of cells analyzed. Arrows indicate representative cytoplasmic puncta in mutants lacking the indicated early-arriving endocytic protein(s). "*" indicates that Ede1 cortical patches are still visible at cell cortex. The scale bar is 5 μm.

mutated in the same protein to express an ede1 EH*-Δsyp1-binding-GFP double mutant, this protein was only detected in cytoplasmic puncta (Fig 7D and 7E). Similarly, the vast majority of ede1 mutant protein was present in cytoplasmic puncta in *syp1 1–565 a.a. ede1 EH** double mutants (Fig 7F) and in *yap1801 Δ yap1802Δ ede1-Δsyp1-binding* triple mutants (Fig 7G). Moreover, Ede1 was mostly observed in cytoplasmic puncta in *syp1 1-565a.a. yap1802 1-339a. a. yap1801Δ* triple mutants (S3 Fig).

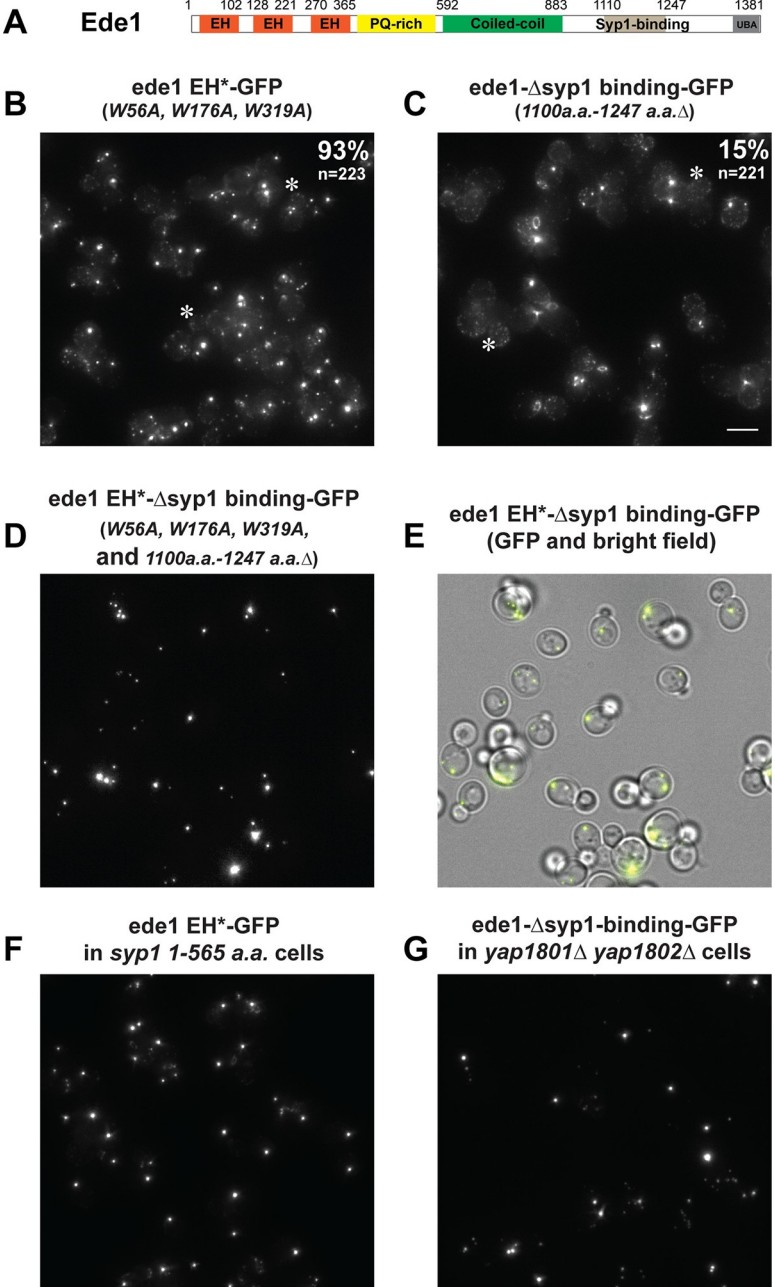

**Fig 7. EH and Syp1 binding domains of Ede1 are required for its cortical localization.** (A) Schematic diagram of Ede1 domain structure. (B–G) Maximum Z-projections of indicated GFP-tagged ede1 mutant proteins in yeast cells. The numbers in the upper right corner of the images in (B) and (C) indicate the percentage of cells that contain cytoplasmic puncta. *n* equals the number of cells analyzed. (E) A merged image of the GFP channel (D) and corresponding brightfield view. Note: the cell outlines are no longer visible in the GFP channel (D) due to the GFP signal concentrating in the cytoplasmic puncta. "*" indicates cells in which cortical patches are still visible in *ede1* mutants. The scale bar is 5 μm.

Our results indicate that Ede1's interactions with the Yap180s, Pals (through EH domains), and Syp1 (through Syp1 binding domain) are crucial for Ede1 recruitment to the cell cortex. In contrast, as we showed earlier, polarized cortical localization of Yap180s or Syp1 is

independent of Ede1. Our data suggest that polarized cortical recruitment of both the Yap180s and Syp1 triggers polarized Eps15 (Ede1)-centric CME initiation complex assembly.

## Syp1 and Yap180s function in a concerted manner to recruit Ede1

We sought to examine further the ability of Yap1802 and/or Syp1 to trigger Ede1 assembly in vivo. Pil1 is a key component of immobile cortical puncta called eisosomes [65], which have distinct localization from CME sites (Fig 8A) [66]. In small-budded yeast cells, eisosomes form exclusively in mother cells, not in daughter cells (Fig 8A, upper right panel) [67]. Thus, the eisosome localization pattern is the inverse of the CME site localization pattern in polarized yeast cells undergoing bud growth. We utilized the rapamycin-induced FRB-FKBP dimerization system [6] to artificially direct FRB-tagged Yap1802 and/or FRB-tagged Syp1 to eisosomes (FKBP12-tagged Pil1) and examined if and how artificially concentrated Yap180s and/or Syp1 at the mother cell cortex may affect Ede1 assembly.

Cells carrying Pil1-TagBFP2-FKBP12, Ede1-mScarlet-I, Yap1802-GFP-FRB, and Syp1-FRB (strain YSY5502) were treated (Fig 8B) or not treated (Fig 8A) with rapamycin, fixed, and Z-stacks were collected using a confocal microscope with Airyscan 2.0 detection. Each fluorescently labeled protein was expressed from its own promoter at its native chromosomal locus to minimize dosage-induced functional or localization perturbation.

In the absence of rapamycin, Yap1802-GFP-FRB and Pil1-TagBFP2-FKBP12 localized exclusively in daughter or mother cells of the YSY5502 strain, respectively (Fig 8A, lower right panel). This result further supports our earlier finding that Yap1802 mostly localizes to daughter cells (Figs 3E–3G and S1). While some Ede1-mScarlet-I (presumably with Syp1-FRB) sites were visible in mother cells (Fig 8A), consistent with a previous report, Ede1-mScarlet-I sites were distinct from eisosomes labeled by Pil1-TagBFP2-FKBP12 (Fig 8A) [66]. Quantitatively, the average Pearson correlation coefficient (PCC) between either Pil1-TagBFP2-FKBP12 and Yap1802-GFP-FRB or Pil1-TagBFP2-FKBP12 and Ede1-mScarlet-I in the rapamycin untreated YSY5502 cells was close to 0.2, indicating negligible correlation (Fig 8E and 8F). After 10 min of rapamycin treatment, Yap1802-GFP-FRB (and presumably Syp1-FRB) were mostly directed to Pil1-TagBFP2-FKBP12-labeled eisosome in the mother of YSY5502 cells (Fig 8B). More importantly, Ede1-mScarlet-I also appeared on Pil1-TagBFP2-FKBP12 puncta in the rapamycin-treated YSY5502 cells (Fig 8B). Consequently, the Yap1802-GFP-FRB and Ede1-mScarlet-I cortical signals in daughter cells were reduced (Fig 8B). In the rapamycin-treated YSY5502 cells, the average PCC between Pil1-TagBFP2-FKBP12 and Yap1802-GFP-FRB (Fig 8E) or between Pil1-TagBFP2-FKBP12 and Ede1-mScarlet (Fig 8F) was 0.68 or 0.63, respectively, indicating high degree of co-localization. Thus, artificial recruitment of Yap1802 and Syp1 to eisosomes effectively redirects Ede1 to the mother cell, where the eisosomes are located.

The effect of Yap1802 or Syp1 alone on Ede1 recruitment to eisosomes was also examined (Figs 8C, 8D and S4). The YSY5455 or YSY5458 strains carry either Yap1802-GFP-FRB or Syp1-GFP-FRB, respectively, and carry both Pil1-TagBFP2-FKBP12 and Ede1-mScarlet-I. Like in the YSY5502 strain, rapamycin effectively directed Yap1802-GFP-FRB or Syp1-GFP-FRB to Pil1-TagBFP2-FKBP12 puncta in the YSY5455 strain or in the YSY5458 stain, respectively, with high PCC scores (approximately 0.7) (Figs 8E and S4). However, the average PCC between Pil1-TagBFP2-FKBP12 and Ede1-mScarlet in YSY5458 strain or in YSY5455 strain was 0.30 or 0.35, respectively, indicating weak co-localization (Fig 8F). These results show that artificially directing Syp1 alone or Yap1802 alone to eisosomes can only trigger modest Ede1 recruitment.

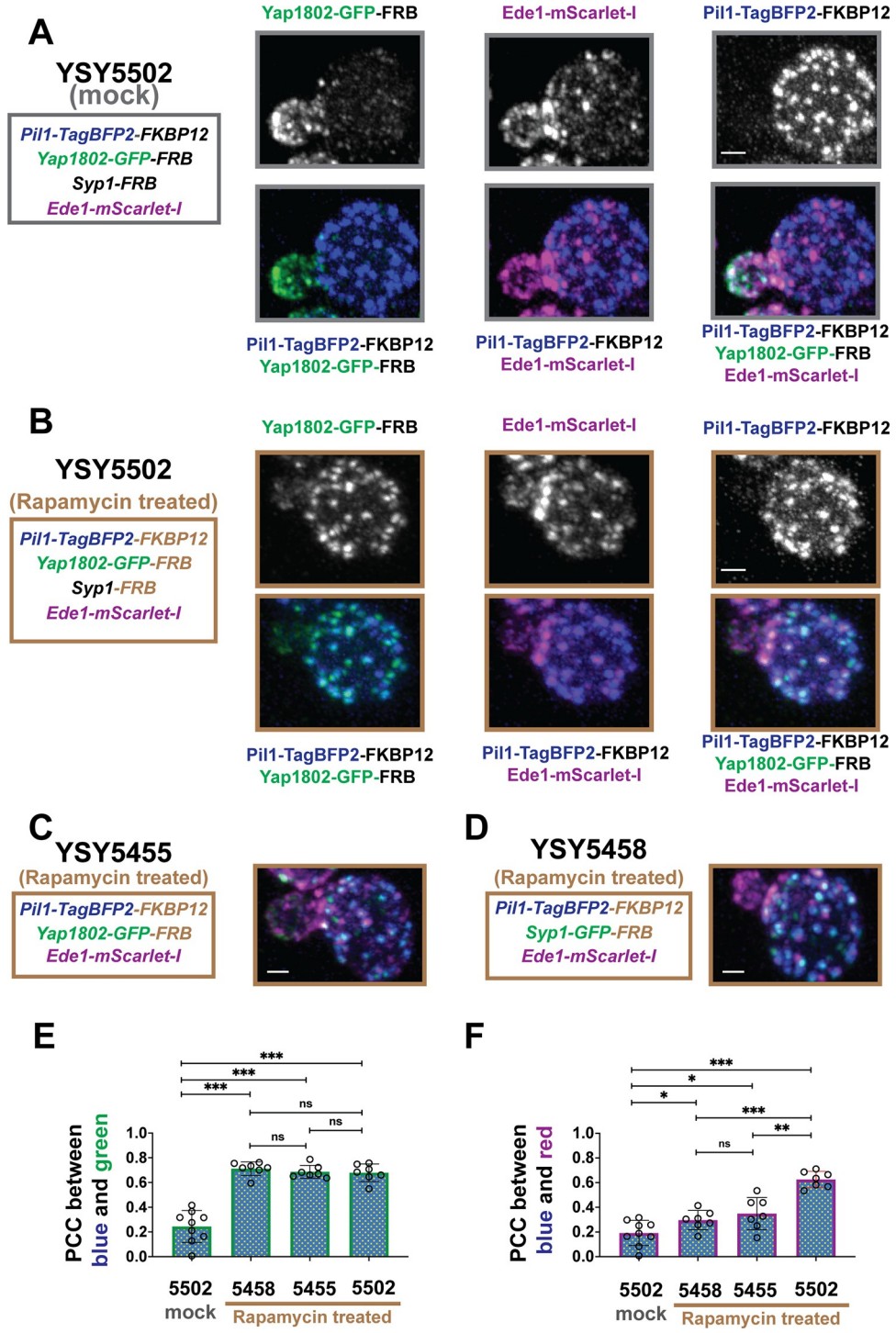

**Fig 8. Syp1 and Yap1802 function in concert to guide CME site initiation through Ede1 recruitment.** (A) In the absence of rapamycin, Yap1802-GFP-FRB and Pil1-TagBFP2-FKBP12 exclusively localize either to daughter cells or mother cells, respectively. Ede1-mScarlet-I patches in mother cells do not colocalize with Pil1-TagBFP2-FKBP12. (B) After treatment with 10 μm rapamycin for 10 min, Yap1802-GFP-FRB (and presumably Syp1-FRB) efficiently relocates to Pil1-TagBFP2-FKBP12 puncta in the mother cell through the FKBP12-rapamycin-FRB interaction, triggering Ede1-mScarlet-I to form patches on Pil1-TagBFP2-FKBP12 puncta in the mother cell. (C, D) Ede1-mScarlet-I localization after either Yap1802-GFP-FRB or Syp1-GFP-FRB alone was targeted to Pil1-TagBFP2-FKBP12 puncta. All data shown are Maximum Z-projections generated from 3-color Z-stacks generated from Airyscan fixed cell imaging. The numerical data are presented in S5 Data. The scale bar is 1 μm. CME, clathrin-mediated endocytosis.

Taken together, our results support the conclusion that Syp1 and Yap1802 function concertedly to mediate optimal Ede1 recruitment.

## Discussion

### Comparative analysis of key endocytic proteins in mother versus daughter cells provides insights into the mechanism of CME site initiation and maturation

CME events were previously shown to have longer lifetimes in mother cells than in daughter cells within yeast undergoing polarized growth [68]. Our recent study further proposed that CME sites mature slowly in the mother cells, likely due to lower cargo levels [56]. These previous observations led us to thoroughly compare CME protein dynamics between mother and daughter cells, and to exploit the intrinsic polarity of budding yeast cells to test hypotheses concerning CME initiation. To better examine how CME proteins and CME sites are influenced by endocytic cargo and/or anionic phospholipids, we developed a strategy to automatically generate separate circumferential kymographs of the 2 compartments of a polarized budding yeast cell, mother and daughter cells, which, respectively, carry reduced and elevated exocytosis and anionic phospholipid levels. These circumferential kymographs provide a visually intuitive 2D representation of all endocytic events occurring at the equatorial focal plane of mother or daughter cells over the duration of a movie, quantitatively and qualitatively enhancing our analysis and generating new insights into CME site initiation and maturation (summarized in Fig 4).

Our comparative analysis revealed that, despite sharing a common pool of endocytic proteins, CME events are initiated much more frequently in daughter versus mother cells (Figs 2C and 3D), indicating that CME site initiation rates are much higher on the daughter cell plasma membrane compared to the mother cell plasma membrane. In addition, the lifetimes of most early- and mid-arriving CME proteins are rather short and regular in the daughter cells (Fig 4B). Consistent with our previous study [56], the lifetimes of Ede1 and Sla2 are long and irregular in mother cells (Fig 4B). We also found that the lifetime of late-arriving endocytic actin machinery is similar between the mother and daughter cells of polarized yeast (Fig 4B). Thus, our results indicated that CME sites mature slower on the mother cell plasma membrane compared to the daughter cell plasma membrane. Together, our new data strongly support the conclusion that the asymmetric distribution of actin patches observed in polarized yeast is caused not only by slower CME site maturation rates in mother cells [56], but also by higher site initiation rates in daughter cells.

Another new observation is that not every CME site has the same protein composition. Yap1802 is mostly absent from CME sites in mother cells of yeast undergoing polarized daughter cell growth (Figs 3G and S1). Similarly, not every CME site in mother cells contains Pal1 or AP-2 (S1 Fig). Lower levels of Yap180s, Pals, and AP-2 in mother cells correlate well with the much lower frequency of CME site initiation, consistent with the idea that these early endocytic proteins play roles in determining the frequency of CME site initiation.

Pan1 (yeast intersectin) lifetime was previously thought to be short irrespective of the location of the CME site within a yeast cell (Fig 1A, left panel). However, here we observed that the Pan1 lifetimes in mother cells are more than twice as long as in daughter cells (Fig 2E). Additionally, we observed a previously missed, low-abundance Pan1 recruitment stage, which is followed by a high-abundance stage (Fig 2D). Intriguingly, the length of Pan1's high-abundance phase in mother cells is very similar to Pan1's lifetime in daughter cells. Thus, the slow and irregular CME site maturation in mother cells correlates with Pan1's longer lifetime in mother cells and its recruitment in 2 stages. We speculate that the transition in Pan1's recruitment

from low to high levels might serve as a signal to trigger CME site maturation, consistent with Pan1 playing a crucial role in linking actin assembly to CME sites during the latter part of the CME internalization [50–52]. Future studies should address if and how lipid and/or cargo might affect Pan1 recruitment and influence CME site maturation.

## Yeast CALM/AP180 and FCHo coordinate with anionic phospholipids and possibly cargo to promote yeast Eps15 assembly and CME site initiation

Several well-conserved endocytic adaptor proteins, including AP-2, FCHo1/2 (Syp1), Eps15/ Eps15R (Ede1), and CALM (YP180s) appear at CME sites at the earliest stages of CME site formation in both mammalian cells and yeast [3,15]. These proteins were known to interact with each other and some carry motifs that bind to cargos and/or anionic lipids such as PI(4,5)P$_2$. These interactions may function redundantly to achieve a robust and flexible CME site initiation [6,24]. In addition, some early-arriving proteins interact with later-arriving proteins and therefore are thought to play additional roles at the later stages of CME site maturation. Due to this complexity, the exact mechanism by which anionic lipids and/or cargo facilitate CME site initiation has remained elusive. However, with intrinsic polarity of molecules implicated in CME initiation, powerful genetics, and robust image-based analytical tools, budding yeast provide a unique opportunity to investigate the molecular mechanisms of CME site initiation in vivo.

The early arriving CME proteins, Yap180s, Pals, and Syp1 maintain polarized cortical localization in daughter cells even in the absence of Ede1 (Figs 5 and S2). Furthermore, the N-terminal half of Yap180s and Syp1, which contain lipid-binding domains but lack Ede1-binding domains, are necessary (S2 Fig) [12] and sufficient (Fig 5B and 5C) for their polarized cortical localization. This observation implicates polarized anionic phospholipid distribution in polarized cortical recruitment of Yap180s and Syp1 to daughter cells. Yap180s are adaptors for CME of the vesicle-associated SNARE protein Snc1 [54]. A recent structural study revealed that a unique C-terminal region within the ANTH domain of Yap1801/2 binds to ubiquitin, which often serves as a cargo internalization signal [59]. Indeed, point-mutation of the cargo-binding site within Yap1802's ANTH domain alone reduced its polarized cortical localization. Syp1 interacts with cargo including Snc1, Mid2, Ptr2, and Mep3 through DxY motifs [69]. Thus, polarized distribution of cargo molecules is implicated in recruiting and/or stabilizing the Yap180s and Syp1 to the cortex of daughter cells.

In contrast to the lack of an effect of Ede1 absence on polarized cortical recruitment of Yap180s and Syp1, we observed that Ede1 accumulates in cytoplasmic puncta in 11 mutants in which various early-arriving proteins are absent or partially truncated (Figs 6 and S3). Moreover, the phenotypes of *ede1* alleles in which only the EH domain and/or the Syp1-binding domain was mutated (Fig 7) strongly support the conclusion that Ede1 is recruited to the cell cortex via interaction of its EH domains with NPF motifs of the Yap180s and Pals as well as through its interaction with Syp1. These results lead us to conclude that Yap180s and Syp1 play key roles in Ede1's cortical recruitment (Fig 9A). Fluorescence (cross-)correlation spectroscopy revealed interactions between Ede1 and itself and Syp1 in the cytoplasm [55]. Ede1 is therefore likely to form a complex with Syp1 in the cytoplasm before these proteins associate with the plasma membrane (Fig 9A). No cytoplasmic interaction was detected between Ede1 and the Yap180s [55], implying that the Ede1-Yap180s interaction through EH domains and NPF motifs only occurs on the plasma membrane (Fig 9A). Establishment of Ede1 interactions with Syp1 and Yap180s in the cytoplasm and at the plasma membrane, respectively, is consistent with our conclusion that Syp1 and the Yap180s contribute to Ede1 recruitment to the plasma membrane in a concerted manner (Figs 6–8).

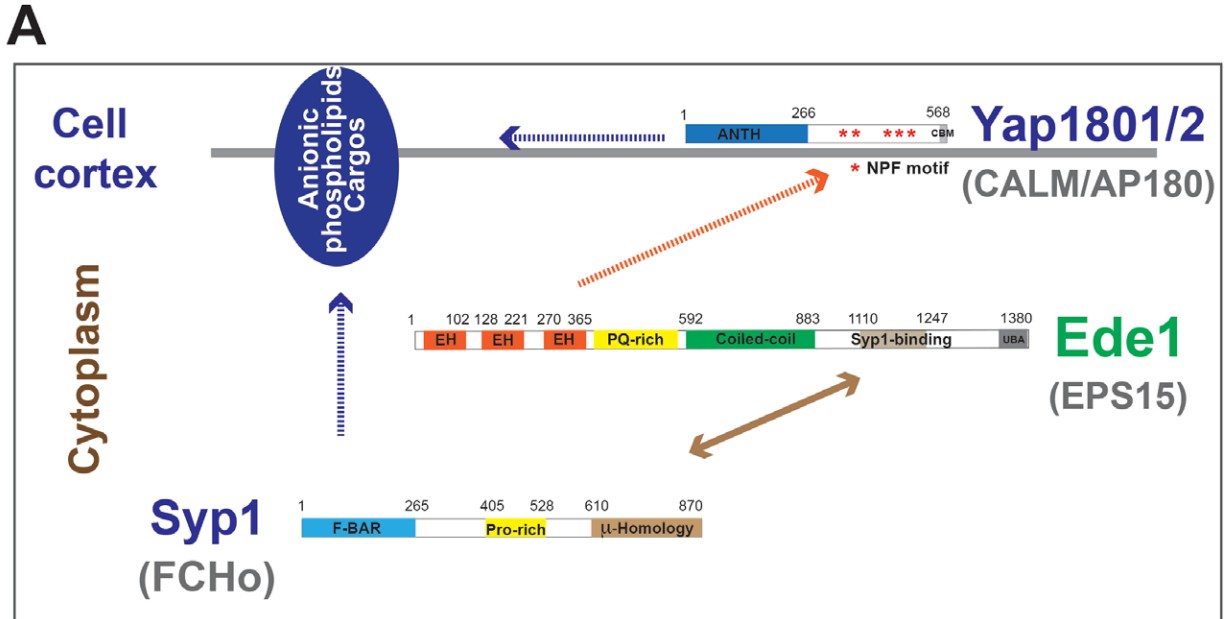

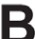

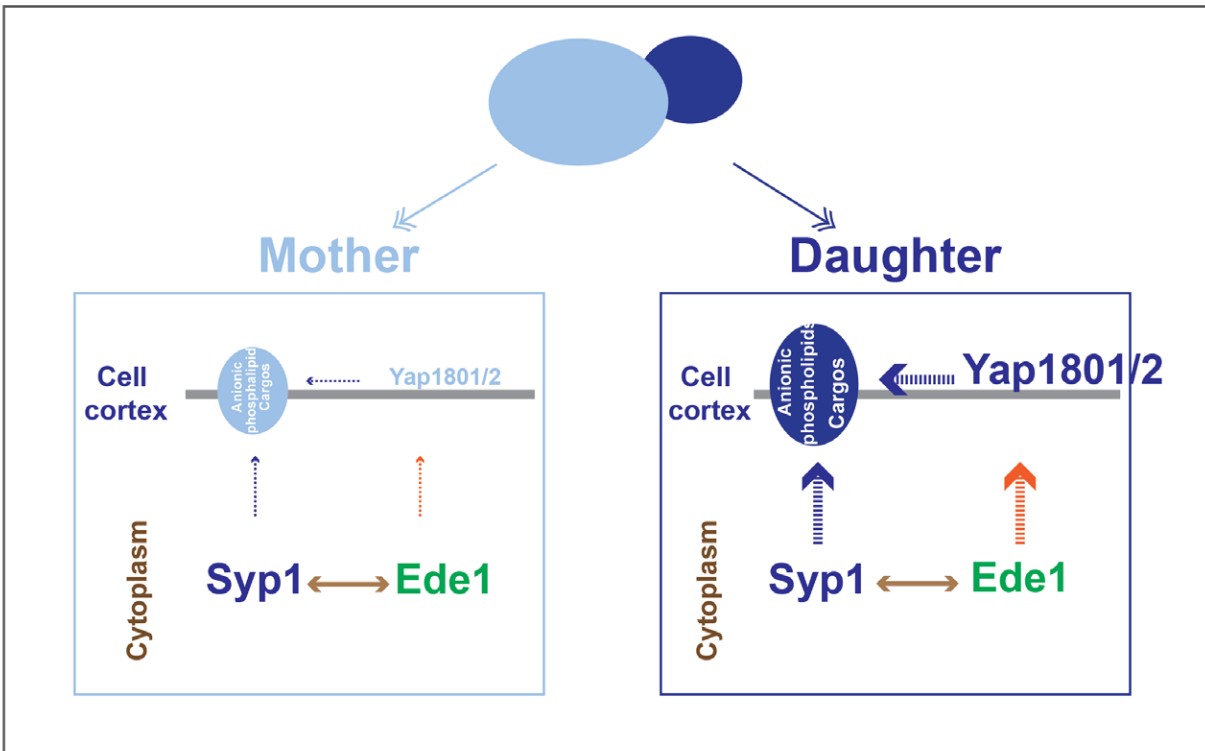

**Fig 9. The molecular mechanism of CME site initiation.** (A) A model in which the Yap180s and Syp1 play major roles in CME site establishment through their interaction with lipids, cargo, and Ede1. The Yap180s and Syp1 are recruited to the plasma membrane (gray line) through interactions with anionic phospholipids and possibly cargos on the plasma membrane. Ede1 forms a complex with Syp1 in the cytoplasm, and this complex is recruited to the plasma membrane while the Ede1-Yap180s interactions occur at the plasma membrane. Mammalian homologs of indicated yeast proteins are shown in gray. Dotted arrow lines: interaction at the plasma membrane. Two-way solid

arrow line: interaction in cytoplasm. (B) A model for polarized CME site initiation in budding yeast. Anionic phospholipids and cargo molecules are enriched at the plasma membrane of daughter cells, where they promote recruitment of the Yap180s and Syp1. Yap180s and Syp1 ensure that Ede1-driven CME site formation occurs where cargo levels are high and acidic phospholipids are abundant. CME, clathrin-mediated endocytosis.

The supportive roles of AP-2, Pal1, and Pal2 in Ede1 recruitment are revealed when Syp1 and/or the Yap180s are absent (Figs 6 and S3). These results are consistent with the hypothesis that early-arriving proteins act in different combinations in response to various cargos at different local environments. Compared to AP-2 in yeast [70], mammalian AP-2 plays a more critical role in CME [3,71], although its importance in CME site initiation is still unsettled [4,5]. Unlike the Yap180s and Syp1, AP-2 and Ede1 show interdependence in their association with the cell cortex (S2 Fig). Thus, AP-2 might affect Ede1 localization in a different way from the Yap180s and Syp1. While NPF-containing Pal1 and Pal2 are conserved across fungi [72], no clear mammalian homolog for these proteins has been identified. Nonetheless, over 70 proteins containing at least 2 NPF motifs in humans [73] are potential candidates to serve similar functions to Pals.

## A molecular mechanism for flexible and robust CME site initiation

Based on our findings and previous knowledge, we propose that Yap180s and Syp1 play central roles in CME site initiation by coordinating interactions with phospholipids, cargo and Ede1 (Fig 9). In this proposed mechanism, the Yap180s, as well as Syp1, play the primary regulatory role by being stably recruited to the plasma membrane in response to the detection of both lipids and/or cargos to ensure coordinated cargo loading and Ede1-driven CME site initiation. Consistently, the mother cell cortex, with lower levels of anionic phospholipids and cargo, shows markedly lower Yap1802 (and Pal1) recruitment levels and fewer CME sites (Fig 9B). In addition, the slower CME site maturation observed in the mother cells (Figs 3 and 4) is in agreement with the hypothesis that cargo plays a role in CME site maturation [56].

A previous study demonstrated that assembly of middle- and late-arriving proteins still occurs to some extent in cells lacking seven early arriving proteins, Ede1, Syp1, Yap1801, Yap1802, Pal1, Pal2, and an AP-2 subunit (7Δ mutant) [6]. Here, we observed that in wild-type cells, some endocytic sites at the mother cell cortex do not contain detectable Yap1802 and Pal1, yet undergo CME, though at a reduced rate. These results support the notion that CME site initiation does not occur in an invariant manner but exhibits a high degree of plasticity. Similar to Yap180s, Sla2 contains an ANTH domain, which binds to PI(4,5)P$_2$ and ubiquitin, which is a cargo internalization signal [59]. NPF motifs in Ent1/2 (yeast Epsins) interact with EH domains in Pan1 [74]. In addition, the ENTH (Epsin N-terminal homology) domain and UIMs (ubiquitin-interacting motifs) of Ent1/2 interact with PI(4,5)P$_2$ and ubiquitin [74]. Thus, while the Ede1-centric early-arriving protein interaction network is lacking in 7Δ cells, the middle-arriving proteins Sla2, Ent1/2, and Pan1 presumably can still partially fulfill the domain requirements proposed in our model, mediating interactions with anionic phospholipids and cargos on the plasma membrane to initiate assembly of CME sites.

In total, our model supports the notion that CME site initiation is not determined by one key factor. Instead, lipids, cargo, and several shared functional domains contained within several multivalent CME proteins likely interact with each other in various combinations to promote CME site formation to accommodate diverse environments (including those present in daughter versus mother cells).

### Relationship of Yap180/CALM to the proposed formation of cortical Ede1/Eps15 liquid-like condensates

Recent studies provide evidence that Ede1 in yeast [19], Eps15 in mammals [18], and AtEH1 (partial homolog of Eps15) in plants [21] have the capacity to form liquid-like condensates and promote CME site assembly. The molecular mechanism of cortical Ede1/Eps15/AtEH1 condensate formation is unclear. Previous results suggest that $PI(4,5)P_2$-interacting FCHo in mammals or PA-PI(4)P-interacting AtEH1 in plants contributes to Eps15 or AtEH1 cortical recruitment, respectively [18,21]. Here, in addition to Syp1 (yeast FCHo), our results revealed that EH-NPF domain interactions between Ede1 and the Yap180s (and the Pals) play important roles in constraining Ede1 assembly at the yeast plasma membrane. Ede1 cytoplasmic puncta were observed more frequently in *yap1801Δ yap1802Δ* or *yap1801Δ yap1802Δ pal1Δ pal2* cells than in *syp1Δ* cells (Figs 6C, 6D and S3). The Yap180s and Pals, carrying 5 or 2 NPF motifs, respectively, and Ede1, carrying 3 EH domains, are ideal partners for forming a multivalent interaction network and promoting liquid–liquid phase separation. While Ede1 forms a complex with Syp1 in the cytoplasm, Ede1 only interacts with the Yap180s at the cell cortex, potentially preventing cytoplasmic Ede1 assembly. Thus, our study has important implications for understanding Ede1-centric condensate formation.

Homologs of the Yap180s are present and well conserved in various organisms, and are known as CALM and AP180 in mammals [75]. A subset of CALM and AP180 homologs has NPF motifs, which bind to EH domains [9,76]. The ANTH domain within CALM interacts with $PI(4,5)P_2$ and cargos [59,77]. Indeed, CALM is one of the most abundant clathrin adaptors (equally in abundance to AP-2) [78] and is proposed to be a major factor in determining the rate of endocytic cargo internalization by controlling CCV size and maturation rate [79–81]. Determining whether and how mammal CALM and AP180 function in Eps15 (condensation)-driven CME site initiation will be important goals for future studies.

## Materials and methods

### Media and strains

Yeast strains were grown in standard rich media (YPD) or synthetic media (SD) supplemented with the appropriate amino acids. The yeast strains used in this study are listed in S1 Table. GFP, mCherry, mScarlet-I, GFP-FRB, and TagBFP2-FKBP12 tags were endogenously integrated at the C-terminus of each gene as described previously [82,83].

### Fluorescence microscopy

Live-cell imaging was performed using a Nikon Eclipse Ti microscope (Nikon Instruments, Melville, New York, United States of America) controlled by Metamorph (Molecular Devices, Sunnyvale, California, USA), equipped with a Plan Apo VC 100×/1.4 Oil OFN25 DIC N2 objective (with Type NF immersion oil, Nikon), a Perfect Focus System (Nikon), and a Neo sCMOS camera (Andor Technology, South Windsor, Connecticut, USA) (65 nm effective pixel size). Cells were grown to early log phase at 25˚C. The cells in synthetic media were adhered to the surface of a concanavalin A coated (0.1 µg/ml) coverslip. All imaging was done at 25˚C. For single-channel, live-cell imaging, images were acquired continuously at 1 frames/sec. Two-channel movies were made using the SPECTRA X Light Engine (Lumencor, Beaverton, Oregon, USA) for excitation with a 524/628 nm dual-band bandpass filter for GFP/mCherry emission (Brightline, Semrock, Lake Forest, Illinois, USA). The time to acquire 1 image pair is 1 s. The 3D stacks were acquired with 0.15 µm vertical spacing.

Airyscan imaging for fixed cells: Cells in synthetic media were adhered to the surface of a concanavalin A coated (0.1 μg/ml) coverslip. The cells were treated with 10 μm of Rapamycin in DMSO or just DMSO for 10 min before fixation. Paraformaldehyde was directly added to the culture to a 4% final concentration, followed by 15-min incubation. Cells were washed twice with CS (cytoskeleton) buffer (10 mM MES, 150 mM NaCl, 5 mM EGTA, 5 mM Glucose, 5 mM MgCl$_2$, 0.005% NaN3, pH 6.1) containing 50 mM NH$_4$Cl, for 5 min with gentle shaking and then 3 times with CS buffer for 5 min. Fixed samples were imaged on a Zeiss LSM900 with Airyscan 2.0 detection. The system was preincubated at 25˚C for 1 h before imaging. Images were acquired in SR-2Y mode, 2.5× Zoom, unidirectional scan, with the max laser scan speed. All channels for a slice of the z-stack were acquired before moving to the next slice. The Z-stack interval was set to "optimal" (0.130 μm). Samples were imaged with 1% laser power, 850V gain, and 1.0 digital gain. The emission filter settings were: RFP 545 to 620 nm, GFP 490 to 560 nm, and BFP 400 to 495 nm. Post-acquisition, images were processed with 3D auto processing for visual representation and 3D processed with SR 9.0 setting for colocalization analysis. Pixel offset was corrected using the Zen blue software's channel alignment (extended) function, using PSF obtained from a z-stack image of 0.2 μm TetraSpec beads as a reference.

## Image and data analysis

ImageJ software was used for the general processing of images and movies, such as background subtraction and photobleaching correction [84].

Circumferential kymographs were produced using the five-step workflow shown in Fig 1. First, images were blurred to reduce background noise and improve signal. Cells were then segmented using pre-trained models in Cellpose 1.0, a deep-learning segmentation algorithm; produced segmentation masks were saved as.tif images. Masks were converted to labeled ImageJ ROIs using a custom-written plugin. As the bud-neck region contained excessive fluorescence unsuitable for kymograph analysis, bud-neck regions were removed from ROIs during the conversion process. ROI widths were manually tuned to cell membrane thicknesses as viewed on each movie's maximum intensity projection. Finally, kymographs were generated using the cmeSpot plugin [47]. The number of endocytic events was counted, and the lifetime was quantified by measuring the length of endocytic events represented on the kymographs [43]. The *p* value was calculated by the Mann–Whitney test in the Prism 10 program. This workflow has been implemented as a software library using Jupyter Notebook and ImageJ user interfaces and is open source under a Berkeley Software Distribution (BSD) 3-clause license and can be found on Github: https://github.com/geanhu/cme-movie-analysis-2024.

To compare the cortical fluorescence signal intensity of GFP-tagged endocytic proteins between yeast mother and daughter cells undergoing polarized growth, a maximum intensity projection of the indicated movie was first produced by the workflow described above. The average intensity along the cell cortex of the mother or the daughter was measured using ImageJ software (https://imagej.nih.gov/ij/). The ratio (daughter versus mother) of the average intensity was calculated for each polarized cell. At least 10 cells were analyzed for each yeast strain, and the average was calculated and presented.

Maximum Z-projection images of cells containing cytoplasmic wild-type or mutant Ede1 puncta in various strains were visually inspected and quantified. Cells containing at least 1 visible cytoplasmic spot were classified as puncta positive cells.

To quantify colocalization of Pil1-TagBFP2-KBP12 with various proteins, 2-color Maximum Z-projection images were analyzed by JACoP plugin [85] in ImageJ. The Pearson's coefficient correlation of indicated protein pairs in Fig 8 was determined by JACoP plugin and

plotted using Prism 10 program. The *p* value was calculated by Mann–Whitney test in Prism 10 program.

## Supporting information

**S1 Fig.** (A) Single images of cells endogenously expressing Yap1801-GFP or Yap1802-GFP. The 2 images were created using identical microscopy settings. The bar is 5 μm. (B) Two-color circumferential kymograph representations of mother and daughter cells endogenously expressing Apl1-GFP/Myo5-mScarlet-I or Pal1-GFP/Myo5-mScarlet-I.
(TIF)

**S2 Fig.** (A, B) Localization of Apl1 (A) or Pal1 (B) in wild-type or *ede1Δ* cells. Maximum intensity projections (MIP) of 3 min-movies were created for the indicated yeast strains. The arrows highlight the polarized localization of Apl1-GFP (A, left panel) and Pal1-GFP (B, left panel) in wild-type cells. The arrows indicate that polarized Apl1-GFP localization is lost (A, right panel) while the polarized Pal1-GFP localization (B, left panel) is retained in *ede1Δ* cells. (C). Localization of ANTH domain truncated-yap1802. The arrows indicate that yap1802 Δ 1–266 a.a.-GFP no longer localizes at the cortex of the daughter cell. The scale bar is 5 μm.
(TIF)

**S3 Fig. Cytoplasmic Ede1 puncta in the indicated mutants.** 3D Z-projection of Ede1-GFP (A) or Ede1-Scarlet-I (B) localization in the indicated mutants. The scale bar is 5 μm.
(TIF)

**S4 Fig. Ede1-mScarlet-I localization after either Yap1802-GFP-FRB or Syp1-GFP-FRB alone was recruited to eisosomes (Pil1-TagBFP2-FKBP12 puncta).** (A) or (B) represents detailed data for Fig 8C or 8D, respectively.
(TIF)

**S1 Table. Yeast strains were used in this study.**
(PDF)

**S1 Data. The numerical data for figure panels 2B, 2C, 2E, 2F.**
(XLSX)

**S2 Data. The numerical data for figure panel 3D.**
(XLSX)

**S3 Data. The numerical data for figure panel 4B.**
(XLSX)

**S4 Data. The numerical data for figure panels 5B and 5C.**
(XLSX)

**S5 Data. The numerical data for figure panels 8E and 8F.**
(XLSX)

## Acknowledgments

We thank members of the Drubin/Barnes laboratory for helpful discussions. We are grateful to Dr. Jonathan Wong and Dr. Meiyan Jin for critically reading our manuscript and for their constructive suggestions. We are grateful to Paul Marchando for providing the *tor1-1* mutant strain.

## Author Contributions

**Conceptualization:** Yidi Sun.

**Data curation:** Yidi Sun.

**Formal analysis:** Yidi Sun.

**Funding acquisition:** David G. Drubin.

**Investigation:** Yidi Sun, Albert Yeam, Yuichiro Iwamoto.

**Methodology:** Yidi Sun.

**Resources:** Albert Yeam, David G. Drubin.

**Software:** Jonathan Kuo, Yuichiro Iwamoto, Gean Hu.

**Supervision:** David G. Drubin.

**Validation:** Yidi Sun.

**Visualization:** Yidi Sun, Yuichiro Iwamoto.

**Writing – original draft:** Yidi Sun.

**Writing – review & editing:** David G. Drubin.

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
