## [Editor Report · Decision Letter 0]

1 May 2024

Dear Dr Drubin, 

Thank you for submitting your manuscript entitled "Yeast CALM/AP180 and FCHo1/2 homologues cooperatively recruit yeast Eps15 to promote clathrin-mediated endocytosis initiation" for consideration as a Research Article by PLOS Biology.

Your manuscript has now been evaluated by the PLOS Biology editorial staff as well as by an academic editor with relevant expertise and I am writing to let you know that we would like to send your submission out for external peer review.

Once your full submission is complete, your paper will undergo a series of checks in preparation for peer review. After your manuscript has passed the checks it will be sent out for review. To provide the metadata for your submission, please Login to Editorial Manager (https://www.editorialmanager.com/pbiology) within two working days, i.e. by May 03 2024 11:59PM.

Kind regards,

Ines

--

Ines Alvarez-Garcia, PhD

Senior Editor

PLOS Biology

---

## [Decision Letter · Decision Letter 1]

18 Jul 2024

Dear Dr Drubin,

Thank you for your patience while your manuscript entitled "Yeast CALM/AP180 and FCHo1/2 homologues cooperatively recruit yeast Eps15 to promote clathrin-mediated endocytosis initiation" went through peer-review at PLOS Biology. Also, please accept my apologies again for the delay in sending you our decision. Your manuscript has now been evaluated by the PLOS Biology editors, an Academic Editor with relevant expertise, and by three independent reviewers.

The reviews are attached below. As you will see, the reviewers find the conclusions interesting and worth pursuing for publication, but they also raise several points that would strengthen the results further. While Reviewer 1 only asks for several clarifications, Reviewer 2 thinks the conceptual advance in understanding CME is modest and that parts of the discussion are a bit redundant and could be streamlined. This reviewer also asks for a better discussion of the model in comparison with previous ones and adding some missing data. Reviewer 3 has two suggestions for improvement: assessing the temporal recruitment of a cargo molecule and addressing the link with anionic phospholipid.

In light of the reviews, we are pleased to offer you the opportunity to address the comments from the reviewers in a revision that we anticipate should not take you very long. After discussions with the Academic Editor, we think that a stronger link with either cargo or anionic phospholipid would consolidate the story further, but we won't make these requests a requirement for publication. However, the Academic Editor thinks you should quantify the distribution of signal in mother and buds in Figure 5.

Once you submit a revision, we will assess your revised manuscript and your response to the reviewers' comments with our Academic Editor aiming to avoid further rounds of peer-review, although might need to consult with the reviewers, depending on the nature of the revisions.

**IMPORTANT - SUBMITTING YOUR REVISION**

3. Resubmission Checklist

a) *PLOS Data Policy*

b) *Published Peer Review*

Sincerely,

Ines

--

Ines Alvarez-Garcia, PhD

Senior Editor

PLOS Biology

Reviewers' comments

Rev. 1:

Sun and Drubin et al present a clever study in which they have studied the differences in endocytosis between mother and daughter budding yeast cells to gain insight into the impact of different imitator and adapter proteins on endocytosis. The author observe a marked difference in the number of successful endocytic events, with daughter cells have far more such events than mother cells. This differences is ascribed to the increase in negatively charged phospholipids and cargo proteins in developing daughter cells. Using this system, the authors present a quantitative analysis of various components of the core endocytic machinery including ede1, yap180, syp1, myosin, and components of the yeast homolog of AP2. Interestingly, the authors find that yap180 and syp1 work together to recruit ede1 to the membrane surface. Once this recruitment occurs, multivalent interactions between ede1 and other endocytic proteins drives endocytosis forward. The study is rigorously performed, and the authors do a good job of putting their work in the context of open questions and past work. I think the paper is already of the quality level necessary for publication. I don't see the need for additional studies. I have only a few questions for the authors to consider as they prepare their manuscript for publication. These are more for my own curiosity rather than critical points.

1. When the authors mention that the daughter cell has more cargo proteins than the mother cell, why types of cargo molecules do they have in mind? How do these cargo molecules interact with the endocytic machinery? Are they likely to interact directly with yap180 vs. AP2, etc.? If so, how might the interactions impact interpretation of the results? I apologize if I have missed this in the discussion somewhere.

2. The authors finding that Syp1 is important for recruitment of ede1 to the plasma membrane is expected, based on recent and older literature. However, I was a bit surprised by the finding that yap180 recruits ede1. I had always thought of ap180/calm (homologs of yap1) as later arriving components. I believe this impression comes from the 2011 paper by Taylor and Merrifield (PLOS Biology), which puts Eps15/Fcho in the earliest group of proteins to arrive and CALM/Epsin/clathrin/ap2 in the second group, still arriving before dynamin. Are things different in yeast? Has it already been established in yeast that Yap180 arrives early? If so, perhaps this could be clarified in the manuscript (apologize if it's already there and I missed it.)

3. Furthermore, if Yap180 indeed arrives early and is important for recruiting Eps15 (as the authors seem to show), then if may imply that snares are playing some role in initiation of endocytic sites? Specifically, as the authors mentioned, David Owen and others showed that AP180/calm are cargo adaptors for snares. What do the authors think about the implications of that in light of their own findings? Could it be that a requisite local concentration of snares is needed to recruit yap180 before eps15 can be recruited and endocytosis can go forward? Is it know whether there is a local increase in snare's in daughter cells? Could it be that this is one of the enriched cargos in the daughter cells? Would like to know your thoughts.

All in all, this is a nice piece of work that uses a clever approach to get some greater clarity on how endocytosis is initiated in yeast, which raises interesting questions (as noted above) about how it might work in mammals. I recommend publishing the study without requiring revision.

Rev. 2:

In this study, the localization kinetics of several proteins involved in clathrin-mediated endocytosis (CME) was carefully examined in both mother and daughter compartments of early budded Saccharomyces cerevisiae cells. To quantitatively compare mother to daughter compartment protein localization intensities and lifetimes, a new image analysis pipeline was developed. This allowed the understanding of individual protein kinetics to be refined and subtle differences in kinetics and intensities revealed relative to previous descriptions. The analysis also validated that CME events are more abundant and regular in the growing and polarized buds and that not all sites of CME contain the full complement of CME proteins. Finally, the localization dependencies among early arriving proteins were examined leading to the conclusions that Syp1, Yap180s and Pal1 do not depend on Ede1 for polarized membrane localization and that Ede1 depends on multivalent interactions with the aforementioned proteins for its cortical recruitment.

The descriptions of protein localization kinetics presented in this study are impressive and thorough, and the new image analysis pipeline may be useful to other groups studying cortical events and processes. The data are clearly presented, statistically sound, the experiments are logical, and the manuscript is also well-organized and written (with the exception noted below). It is a very solid contribution. However, the conceptual advance in understanding CME appears to be modest. Subtle differences with previous descriptions in localization kinetics of CME proteins have been identified. The work validates that CME is preferential to the growing bud and emphasizes the importance of Ede1 recruitment. Apparently, the details of Ede1 recruitment were not known previously but that Ede1 relies on multivalent interactions does not appear to be a major advance in understanding. I have only a few "major" comments for consideration and several minor.

Major comments:

1. Instead of "yeast" or "budding yeast", please use Saccharomyces cerevisiae throughout the manuscript. For example, it is incorrect to state that Ede1 is the "yeast" name for Eps15. S. cerevisiae is not the only yeast used as a model organism for cell biology studies, not even the only budding yeast, and what is described here for S. cerevisiae (timing, protein concentration, etc. at CME sites) may not be generalizable to other yeast.

2. Portions of the discussion (for example, section beginning line 470) are repetitive with what is written in previous sections. Further attention to reducing repetitive elements and minimizing the re-stating of the results and introduction would improve it.

3. This group has made numerous significant contributions to the understanding of CME. In a previous paper, Pedersen et al., 2020, it is stated "Thus, site maturation, rather than site initiation, accounts for the previously observed polarized distribution of actin patches in this organism." This statement seems to undercut the significance of studying CME initiation steps to explain differential CME in mother and bud compartments, particularly since the work presented here shows no difference in Myo5 or Sla1 kinetics. Thus, it would be helpful to the non-expert if the current work and model were discussed more fully in the context of previous work and models.

4. Syp1 is not included in the frequency or lifetime analyses presented in Figs 3 and 4B although it is included in the cartoons of 1A and 4A, and Figure 3H. It seems it is similar, if not identical to Ede1, and the data are apparently available so that information should be added to the text (frequency) and lifetime of 4B for completeness.

Figure comments:

1. Fig 1A. The paper includes work with Pal1/2 and therefore these should also be included in the timeline despite the fact that they are not apparently evolutionarily conserved.

2. Kymographs would benefit from a more detailed time axis - perhaps ticks at every 30 sec.

3. Colorblind readers would appreciate a different dual color scheme, perhaps green/magenta rather than green/red.

Minor comments:

1. Line 177: duplicates information provided above for endogenous tagging of proteins.

2. Line 183: lifetimes

3. Line 197: Yap180s' GFP.

4. Verb tense is not consistent throughout the results section. Typically, results are presented in the past tense (as in line 145) rather than present tense (as in line 142). See again lines 158-161 for another example of switching verb tense.

5. Line 212: mammalian

6. Line 221: results do not conclude. This sentence needs to be re-worded.

7. Line 277: genotypes need to be italicized.

8. Line 310: ditto to point 7.

9. Line 424: this sentence needs to be re-worded.

10. Line 428: incomplete sentence.

11. Line 242: mentions "over time". Would a single time point not show the same result? What is the advantage of including several time points - signal?

12. Line 262: please specify what previous results the current results are consistent with.

Rev. 3: Vytas Bankaitis – note that this reviewer has signed his review

This MS interrogates the mechanism by which the sites of CME initiation on the plasma membrane are determined. While this issue has been studied for decades, and there is general agreement in many areas, there remain questions/controversies regarding order of assembly, etc - particularly as these relate to the initiation phase of the process. The fact that a number of the proteins involved in the early stages of CME do not have obvious lipid-binding motifs contributes to the uncertainty.

Using the yeast system, the authors take advantage of the intrinsic asymmetry of acidic phospholipid and endocytic cargo molecule content (concentration) in newly formed daughter cells (relative to their mothers during polarized cell growth) to assess the relationship between cargo/anionic phospholipid densities, and the temporal recruitment of initiating proteins, as key properties of CME initiation sites. The authors deploy a powerful combination of quantitative imaging and genetic approaches to these ends. The 2D kymograph approach is novel and yields particularly important spatial and temporal information that is essential to the case being made. The conclusion is that CME initiation and maturation occurs at very different frequencies and rates in daughters vs mothers. The authors extend those conclusions to describe a pathway where Yap1801/1802 and Syp1 cooperate with cargo and lipids to launch yeast Eps15-driven CME assembly at those specific sites.

In summary, this is a clever and well-constructed MS that provides information that will advance our understanding of CME initiation and will do so significantly. The differences, and these are clear differences, in CME maturation and life time in daughter vs mother cells is quite extraordinary. There is little to quibble with regarding this well-presented and well-written story.

Minor suggestions for the authors and editor to consider (not to be taken as criticism) are as follows:

(i) The authors invoke cargo and anionic phospholipids in the intellectual construction they present. While this is certainly justified given what is known about CME, the MS would be strengthened by assessing the temporal recruitment of a cargo molecule. It would seem that the v-SNARES Snc1 or Snc2 would suit as their endocytosis would be required for support of the rapid plasma membrane growth in the newly born daughter. It seems to this reviewer that tracking such a reporter would also provide information as to 'productive' vs 'abortive' CME initiations.

(ii) The authors do not address the anionic phospholipid angle. The reviewer acknowledges that ts mutants in the inositol lipid kinases might be problematic. But, it would seem the FRB/FKBP system might be applied to wipe the PM of PtdIns(4,5)P2 or PtdIns4p. The 2D kymograph system would still permit conclusions even though the 'wipe' would likely be isotropic.

---

## [Editor Report · Decision Letter 2]

5 Sep 2024

Dear Dr Drubin,

Thank you for the submission of your revised Research Article entitled "The conserved protein adaptors CALM/AP180 and FCHo1/2 cooperatively recruit Eps15 to promote the initiation of clathrin-mediated endocytosis in yeast" for publication in PLOS Biology. On behalf of my colleagues and the Academic Editor, Sophie Martin, I am delighted to let you know that we can in principle accept your manuscript for publication, provided you address any remaining formatting and reporting issues. These will be detailed in an email you should receive within 2-3 business days from our colleagues in the journal operations team; no action is required from you until then. Please note that we will not be able to formally accept your manuscript and schedule it for publication until you have completed any requested changes.

PRESS

Sincerely, 

Ines

--

Ines Alvarez-Garcia, PhD,

Senior Editor

PLOS Biology
